# CMA-ES for Post Hoc Ensembling in AutoML:
# A Great Success and Salvageable Failure

**Lennart Purucker**[1]  **Joeran Beel**[1]

[1]University of Siegen

**Abstract**   Many state-of-the-art automated machine learning (AutoML) systems use greedy ensemble selection (GES) by Caruana et al. (2004) to ensemble models found during model selection post hoc. Thereby, boosting predictive performance and likely following Auto-Sklearn 1's insight that alternatives, like stacking or gradient-free numerical optimization, overfit. Overfitting in Auto-Sklearn 1 is much more likely than in other AutoML systems because it uses only low-quality validation data for post hoc ensembling. Therefore, we were motivated to analyze whether Auto-Sklearn 1's insight holds true for systems with higher-quality validation data. Consequently, we compared the performance of covariance matrix adaptation evolution strategy (CMA-ES), state-of-the-art gradient-free numerical optimization, to GES on the 71 classification datasets from the AutoML benchmark for AutoGluon. We found that Auto-Sklearn's insight depends on the chosen metric. For the metric ROC AUC, CMA-ES overfits drastically and is outperformed by GES – statistically significantly for multi-class classification. For the metric balanced accuracy, CMA-ES does not overfit and outperforms GES significantly. Motivated by the successful application of CMA-ES for balanced accuracy, we explored methods to stop CMA-ES from overfitting for ROC AUC. We propose a method to normalize the weights produced by CMA-ES, inspired by GES, that avoids overfitting for CMA-ES and makes CMA-ES perform better than or similar to GES for ROC AUC.

## 1   Introduction

Auto-Sklearn (Feurer et al., 2015) was the first automated machine learning (AutoML) system to discover that building an ensemble of models found during model selection is possible in an efficient manner and superior in predictive performance to the single best model. Afterwards, several other AutoML systems also build an ensemble *post hoc*: AutoGluon (Erickson et al., 2020), Auto-Pytorch (Mendoza et al., 2018; Zimmer et al., 2021), MLJAR (Płońska and Płoński, 2021), and H2O AutoML (LeDell and Poirier, 2020) all implemented *post hoc ensembling*.

Besides H2O AutoML, all of these systems implemented *greedy ensemble selection* (GES) (Caruana et al., 2004, 2006), a greedy search for a weight vector to aggregate the predictions of base models. In AutoML systems, GES is trained using the base models' predictions on the *validation data*, which are computed while evaluating a base model during model selection. The frequent usage of GES likely follows Auto-Sklearn's reported insight that alternatives like *stacking* (Wolpert, 1992) or gradient-free numerical optimization overfit and are more costly than GES.

Auto-Sklearn 1, by default, only has limited validation data for post hoc ensembling, that is, a 33% hold-out split of the training data. We deem this to be low-quality validation data because, depending on the dataset, 33% are not enough instances to avoid overfitting while training GES. Hence, we were motivated to analyze if Auto-Sklearn's insight also holds true for an AutoML system with higher-quality validation data, *e.g.*, AutoGluon with $n$-repeated $k$-fold cross-validation. Moreover, we were motivated to focus on gradient-free numerical optimization instead of stacking. Stacking is generally well-known in ensembling for machine learning and is used by H2O AutoML for post hoc ensembling. In contrast, gradient-free numerical optimization has not been used so far.

Thus, we compare the performance of GES to *covariance matrix adaptation evolution strategy* (CMA-ES) (Hansen and Auger, 2014; Hansen, 2016), state-of-the-art gradient-free numerical optimization (Hansen et al., 2010; Szynkiewicz, 2018; Li et al., 2020). We chose CMA-ES due to its widespread usage in numerical optimization (Li et al., 2020). Moreover, CMA-ES's update is efficient and therefore enables fast training in post hoc ensembling; similar to GES's training. Furthermore, the function evaluation in post hoc ensembling, i.e., calculating the score of aggregated predictions, takes seconds (Feurer et al., 2015). Thus, we disregarded Bayesian optimization, which is appropriate for tasks with expensive function evaluation such as hyperparameter optimization (Lan et al., 2022).

In this study, we aim to boost the predictive performance as much as possible with post hoc ensembling. Note that GES selects a small ensemble, while methods like gradient-free numerical optimization or stacking produce an ensemble that includes all base models. Thus, the inference time and size of the final model are larger for the latter two than for GES.

Our first contribution is an application of CMA-ES for AutoGluon on the 71 classification datasets from the AutoML Benchmark (Gijsbers et al., 2022). Thereby, we show that Auto-Sklearn's insight w.r.t. overfitting of gradient-free numerical optimization depends on the chosen metric. We contradict the insight for the metric *balanced accuracy* by showing that CMA-ES statistically significantly outperforms GES. And we confirm the insight for the metric *ROC AUC* by showing that GES outperforms CMA-ES due to overfitting.

As a follow-up, our second contribution is a method to avoid overfitting for CMA-ES. Motivated by the successful application of CMA-ES for balanced accuracy, we explored methods to stop CMA-ES from overfitting to *salvage* CMA-ES for ROC AUC. We identified the chosen method to normalize the ensemble's prediction probabilities as the key to avoiding overfitting. With this knowledge, we propose a novel normalization method, inspired by GES's implicit constraints during optimization, that makes CMA-ES perform as well as GES and avoids overfitting for ROC AUC. Interestingly, our normalization method also enables us to keep the size of the ensemble small.

Our code and data are publicly available: see Appendix E for details.

## 2 Related Work

Besides Auto-Sklearn 1's (Feurer et al., 2015) statement related to post hoc ensembling, only H2O AutoML names theoretical guarantees (van der Laan et al., 2007) as the reason for using stacking, but does not comment on GES. In general, details about post hoc ensembling in publications about AutoML systems were only a short comment without experiments or a reference to Auto-Sklearn 1 (Feurer et al., 2015; Mendoza et al., 2018; Erickson et al., 2020; LeDell and Poirier, 2020). We are only aware of the work by Purucker and Beel (2022), which proposed a first benchmark and framework for post hoc ensembling. The results in their Appendix also showed that GES can outperform stacking. To the best of our knowledge, no other work on post hoc ensembling for AutoML exists.

CMA-ES was previously applied to machine learning problems like hyperparameter optimization (Nomura et al., 2021; Loshchilov and Hutter, 2016) or feature weighting (Tasci et al., 2018)[1]. However, we found no work that used CMA-ES to directly optimizes the weights of an ensemble. Likewise, we have found no work that applies normalization to the solutions produced by CMA-ES nor comparable machine learning methods that apply normalization in this way to combat overfitting.

## 3 Application of CMA-ES for Post Hoc Ensembling

In our application of CMA-ES for post hoc ensembling, we search for an optimal weight vector $W = (w_1, ..., w_m)$ to aggregate pool $P$ of $m$ base models that minimizes a user-defined loss $L(P, W)$. Thereby, $L$ aggregates the predictions of models in $P$ by taking the $W$-weighted arithmetic mean.

Hence, we employ CMA-ES, as implemented in pycma (Hansen et al., 2019), with default values to find $W$ by minimizing $L$. Following GES's first iteration, we set the initial solution $x_0$ to be the

---

[1]To the best of our knowledge, this work is not available in English. We read a machine-translated version.

weight vector representing the single best model, that is, the weight for the single best model is one while all other models are weighted zero. The initial standard deviation is 0.2 following the intuition that a good weight vector might be close to the initial solution and that the granularity of weights can be small, e.g., between 0 and 1, like in GES.

### 3.1 Experiments: CMA-ES *vs.* GES

We compared CMA-ES to GES w.r.t. ROC AUC following the AutoML Benchmark (Gijsbers et al., 2022). ROC AUC requires prediction probabilities and is independent of a decision threshold that would transform prediction probabilities into labels. We use macro average one-vs-rest ROC AUC for multiclass. We complemented the comparison by also evaluating w.r.t. balanced accuracy, which requires predicted labels and, thus, depends on a decision threshold.

For a threshold-dependent metric, the prediction of CMA-ES is, in our application, the class with the highest value after aggregating the prediction probabilities with the $W$-weighted mean. For a threshold-independent metric, we transform the aggregated probabilities for each instance using the softmax function, *i.e.*, we treat the aggregated probabilities of each class as decision functions and take their softmax. Otherwise, the aggregated probabilities would not represent prediction probabilities, as $W$ can have negative or positive values of any granularity.

To compare the ensembling methods, we obtained base models and their validation data with AutoGluon (Erickson et al., 2020) for each fold of the 71 classification datasets from the AutoML benchmark (AMLB) (Gijsbers et al., 2022) – for both metrics. Then, per fold, we trained the ensemble methods on the validation data, i.e., search for $W$, and scored them on validation and test. The final validation/test score of a method for a dataset is the average over the 10 folds.

Following the AMLB, we ran AutoGluon for 4 hours with 8 cores (AMD EPYC 7452 CPU) and 32 GB of memory. We increased the memory for several datasets to 64 or 128 GB to avoid that insufficient memory made it impossible to produce multiple base models. In the end, AutoGluon produced between 2 and 24 base models, see Appendix F for details per dataset and metric.

We used the same resources and hardware to train and evaluate the ensemble methods. However, instead of training ensemble methods for 4 hours, we follow Auto-Sklearn's default and stop training GES after 50 iterations. This results in $m * 50$ total evaluations of $L$ by GES. Therefore, we terminated CMA-ES after $m * 50$ evaluations of $L$.

We included the single best base model (SingleBest) in the comparison as a baseline. To evaluate the statistical difference between the methods, we perform a Friedman test with a Nemenyi post hoc test ($\alpha = 0.05$), following the AMLB. See Appendix I.1 for more details on the statistical tests.

### 3.2 Results: CMA-ES *vs.* GES

We split the results for binary and multi-class classification in all our evaluations following the AutoML Benchmark (Gijsbers et al., 2022). Figure 1 shows the mean rank and results of the statistical test with critical difference (CD) plots. The Friedman tests were significant in all our experiments. We observe that CMA-ES is statistically significantly better than GES for balanced accuracy but fails to perform similarly well for ROC AUC.

To analyze the impact of overfitting on this outcome, we inspect the change of the mean rank of CMA-ES when switching from validation to test data for both metrics, see Table 1. A detailed overview for all methods can be found in Appendix G.1. While the single best is always ranked last, GES overtakes CMA-ES when switching from validation to test data for ROC AUC. Notably, CMA-ES has a mean rank of almost 1 for validation data in 3 out of 4 cases.

On validation data, GES is only competitive for multi-class ROC AUC, where it has a mean rank of 1.6. Nevertheless, GES has a larger distance to the single best on validation for balanced accuracy than it has for test data with a mean rank of ~2 against the single best's ~3.

In summary, we conclude that Auto-Sklearn's insight w.r.t. overfitting does not generalize to an AutoML system with higher-quality validation data, *i.e.*, AutoGluon, for *balanced accuracy*. In

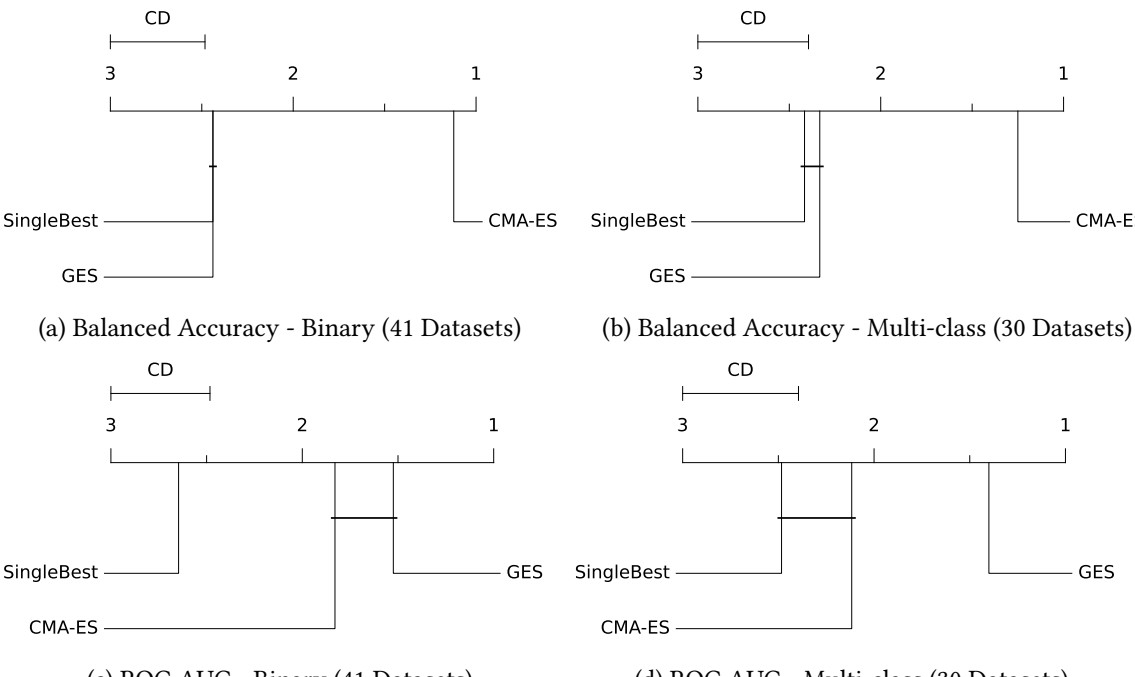

(a) Balanced Accuracy - Binary (41 Datasets)

(b) Balanced Accuracy - Multi-class (30 Datasets)

(c) ROC AUC - Binary (41 Datasets)

(d) ROC AUC - Multi-class (30 Datasets)

Figure 1: **CD Plots Comparing GES and CMA-ES**: The mean rank (lower is better) of a method is its line's position on the axis. Methods connected by a bar are not significantly different.

contrast, *the insight holds for ROC AUC*. Furthermore, we observe that CMA-ES is able to achieve peak performance for ROC AUC on validation data.

## 4 Normalization to Combat Overfitting

The results we just presented motivated us to salvage CMA-ES for ROC AUC. Due to its good performance for ROC AUC and its wide adaptation by AutoML systems, we decided to analyze GES to determine how to avoid overfitting. As a result, we found two properties that inspired our approach to salvage CMA-ES for ROC AUC. This section describes why and how we use normalization to combat overfitting for a threshold-independent metric like ROC AUC. Since our approach is inspired by GES, we start with preliminaries regarding GES and its properties.

### 4.1 Preliminaries

Greedy ensemble selection with replacement (Caruana et al., 2004, 2006) performs an iterative greedy search to build a list of (repeated) base models, the ensemble $E$, that minimizes a user-defined loss function. In each iteration, the base model minimizing the loss, when added to $E$, is *selected* to be part of $E$. To produce predictions and evaluate any $E$, the (repeated) predictions of all base

Table 1: Mean rank change from validation to test data for CMA-ES compared to GES and SingleBest.

| Metric | Task Type | Mean $Rank_{Validation}$ | Mean $Rank_{Test}$ | Absolute Rank (Val → Test) |
|---|---|---|---|---|
| Balanced Accuracy | Binary | 1.00 | 1.12 | 1.0 → 1.0 |
| Balanced Accuracy | Multi-class | 1.03 | 1.25 | 1.0 → 1.0 |
| ROC AUC | Binary | 1.02 | 1.83 | 1.0 → 2.0 |
| ROC AUC | Multi-class | 1.42 | 2.12 | 1.0 → 2.0 |

models in $E$ are aggregated with the arithmetic mean. Taking the arithmetic mean of $E$ weights base models that exit multiple times higher. Hence, given $E$, we can compute a weight vector. Assuming we run GES for $N$ iterations[2], then $|E| = N$ and we compute the weight vector using:

$$W^{pDisc} = \left[ \frac{countIn(p_i, E)}{N} \, \middle| \, p_i \in P \right]. \tag{1}$$

While analysing GES, we found two constraints of the weight vector $W^{pDisc}$ that we believe to be essential for its performance. That is, $W^{pDisc}$ is *pseudo-discrete* and *sparse*. Both properties are only *implicitly* respected by GES and were, to the best of our knowledge, never formally defined.

**Pseudo-Discrete**. We call $W^{pDisc}$ pseudo-discrete because one can transform every weight vector produced by GES into a discrete count of how often a base model has been selected. This can be done by multiplying $W^{pDisc}$ with $N$, reversing Equation 1. In fact, every weight vector produced by GES is in the set $\mathcal{G} = \{W' \mid W' \in H(N) \text{ and } \sum_{i=1}^{m} w_i = 1\}$ with $H(N)$ the $m$-fold Cartesian product of $\{0, 1/N, 2/N, ..., 1\}$:

$$H(N) = \{0, 1/N, 2/N, \dots, 1\} \times \cdots \times \{0, 1/N, 2/N, \dots, 1\}. \tag{2}$$

In other words, every weight $w_i \in W^{pDisc}$ can be expressed as a positive fraction with denominator $N$, and the weight vector sums to 1. This follows from GES iteratively building a list of base models $E$ and calculating the final weight vector with Equation 1.

We would like to remark that this formulation of GES is very similar to mallows' model average (MMA) (Hansen, 2007, 2008; Le and Clarke, 2022) and that GES might share MMA's asymptotic guarantees for regression if $L$ is the squared error (Le and Clarke, 2022).

**Sparse**. $W^{pDisc}$ is sparse, that is, a weight vector where many models are assigned zero weight – as intended for an *ensemble selection* approach (Tsoumakas et al., 2009). To the best of our knowledge, a guarantee for sparseness was never formally introduced or proven for (greedy) ensemble selection, cf. (Caruana et al., 2004, 2006; Tsoumakas et al., 2009). Here, we shortly provide an argument for why it is likely that GES produces a sparse weight vector:

GES only adds new base models to $E$ if they reduce the loss. Hence, it would require at least $m$ iterations where adding a new base model would reduce the loss more than adding an existing base model again (increasing its weight). As a result, for appropriate values for $m$ and $N$, it is unlikely that enough iterations happened such that each model was able to reduce the loss once. Auto-Sklearn, for example, uses $m = 50$ and $N = 50$ by default. Moreover, once $E$ becomes large, the changes to the aggregated prediction that are induced by adding a new base model are minimal. Thus, it also becomes less likely that the changes result in a different loss. Additonally, the larger $E$ is, the more likely GES has reached a (local) optimum, which can not be improved upon by adding new models. In short, the iterative greedy approach to add models to $E$ likely makes $W^{pDisc}$ sparse.

## 4.2 Motivation

Since all solutions produced by GES are pseudo-discrete and (likely) sparse, and since GES does not seem to overfit, we hypothesized that both properties might help to avoid overfitting.

Note, the properties can be seen as constraints. They constrain the weight vector to be sparse, sum to 1, and contain only values such that $0 \le w_i \le 1$. In contrast, our application of CMA-ES uses no such constraints. By default, CMA-ES produces a continuous and dense vector which does not need to sum to 1 and may contain negative or positive values of any granularity.

Thus, our first idea was to constrain the optimization process of CMA-ES such that it would produce results that match the constraints of GES. However, we found that once the same constraints

---

[2]We always denote $N$ as the number of the iteration the final $E$ was found in. Depending on the implementation of GES, the final $E$ does not need to be from the final iteration.

are introduced, CMA-ES often violates the constraints; making CMA-ES inefficient and often leading to an endless loop due to rejection sampling. In other words, we were not able to make CMA-ES produces solution vectors that fulfill all constraints of GES. In general, constraining CMA-ES is also not trivial (Biedrzycki, 2020), and we leave more sophisticated approaches to constrain CMA-ES for post hoc ensembling, like methods based on repair-and-inject or penalization (Hansen, 2016) or with relaxed constraints, to future work.

Instead of constraining the optimization process of CMA-ES, we moved to adding the constraints directly to the weight vector when they are evaluated, following a concept observed from GES. That is, we observed that while the constraints of GES are an implicit result of the algorithm as defined by Caruana et al. (2004), they manifested explicitly only when one computes the weight vector with Equation 1. The optimization loop of GES, *i.e.*, iteratively building $E$, does not explicitly consider these constraints, but only greedily minimizes a user-defined loss. In other words, the optimizer is only implicitly constrained *by applying constraints during the computation of the weight vector; before evaluating the vector's performance.*

In detail, every time GES computes the loss for an ensemble $E$, it first transforms $E$ into $W^{pDisc}$ using Equation 1. Thereby, applying the constraints that the resulting weight vector must sum to 1, is sparse, and $0 \leq w_i \leq 1$. Then, the $L(P, W^{pDisc})$ is returned as the loss of $E$. At this point, it becomes clear that changing Equation 1 leads to different constraints; the loss of $E$ could change without touching the optimization loop of GES.

As a result, we were motivated to apply the same concept to CMA-ES by normalizing the weight vector before we aggregate the predictions of the base models. Thus, changing the loss associated with a weight vector proposed by CMA-ES outside of its optimization process. In contrast, our application in Section 3 normalized the aggregated predictions for ROC AUC using softmax – we normalized *after aggregation*. Now, however, we propose to normalize *before aggregation* as in GES. In turn, this also changes the optimization process of CMA-ES, *e.g.*, the parameter update, because a weight vector might have a different loss depending on normalizing before or after aggregation.

### 4.3 Normalization Methods

We propose three distinct normalization methods. Two of the methods we propose are based on the concept of GES such that the last proposed method tries to simulate Equation 1 fully.

**1) Softmax (CMA-ES-Softmax).** Initially, we propose a simple alternative to our previous usage of CMA-ES by moving the (non-linear) softmax before the aggregation. That is, we normalize the weight vector $W$ by taking its softmax. That is, for a weight $w_i \in W$, we calculate: $w_i^s = \frac{\exp(w_i)}{\sum_{j=1}^{m} \exp(w_j)}$ (3), resulting in $W^s = (w_1^s, ..., w_m^s)$ with $\sum_{j=1}^{m} w_j^s = 1$ and $0 \leq w_j \leq 1$ for $w_j \in W^s$.

**2) Softmax & Implict GES Normalization (CMA-ES-ImplictGES).** Next, we propose to re-normalize $W^s$ with the aim of producing an equivalent to a pseudo-discrete weight vector $W^{pDisc}$; simulating GES's $\mathcal{G}$ (see Equation 2). Therefore, we round each value of $W^s$ to the nearest fraction with denominator $N_{hyp}$ producing a *rounding-discrete* weight vector $W^{rDisc}$. Then, $N_{hyp}$ represents the number of *hypothetical iterations* for a simulated $\mathcal{G}$. We set $N_{hyp} = 50$, similar to GES.

We produce $W^{rDisc} = (w_0^{rDisc}, ..., w_m^{rDisc})$ by multiplying each $w_i^s$ with $N_{hyp}$ and rounding each element to the nearest integer afterwards; rounding up for values larger than 0.5. Therefore, we first compute the integer vector $R = (r_1, ..., r_m)$ using $r_i = \lfloor w_i^s * N_{hyp} \rceil$. Note, $R$ can be thought of as a vector of repetitions where $r_i$ denotes how often a model has been repeated in a hypothetical list of repeated base models $E_{hyp}$. That is, $E_{hyp}$ is connected to $W^{rDisc}$ like an $E$ to its $W^{pDisc}$. Hence, we can compute $W^{rDisc}$ using $R$, paralleling Equation 1:

$$W^{rDisc} = \left\lfloor \frac{r_i}{\sum_{j=1}^{m} r_j} \mid r_i \in R \right\rceil. \tag{4}$$

$W^{rDisc}$ sums to 1, and each element is between 0 and 1. Interestingly, we found that this approach also *implicitly trims* base models, as the nearest fraction can be $\frac{0}{N_{hyp}}$ such that the method assigns zero weight to base models in these cases.

**3) Softmax & Explicit GES Normalization (CMA-ES-ExplicitGES).** Finally, we propose to explicitly trim base models and perfect the simulation of Equation 1. We can explicitly trim base models based on $N_{hyp}$. We found that a weight $w_j^s$ is set to zero by rounding if $w_j^s * N_{hyp} \leq 0.5$. If we reformulate the inequality to $w_j^s \leq 0.5 * \frac{1}{N_{hyp}}$, we see that this parallels GES, where the number of iterations determines the minimal weight a model can be assigned, *i.e.*, $\frac{1}{N}$.

Furthermore, we found that CMA-ES-ImplictGES does not simulate GES sufficiently. We observed that rounding may result in $\sum_{j=1}^{m} r_j \neq N_{hyp}$. That is, the total number of repetitions in $R$ did not match the number of simulated iterations nor the (hypothetical) length of $E_{hyp}$. $R$ was supposed to relate to $E_{hyp}$ for $W^{rDisc}$ like an $E$ to its $W^{pDisc}$. Yet for GES, it holds that $|E| = N$ while $|E_{hyp}| \neq N_{hyp}$ can happen in CMA-ES-ImplictGES.

Considering both, we implemented the third method, shown in Algorithm 1. First, we compute $W^s$ and trim any base model smaller than $\frac{0.5}{N_{hyp}}$ (Line 2). If we set all weights to zero, we fall back to an unweighted average (Line 5). Second, we round to the nearest integer, producing $R'$ (Line 8).

Next, we set $R'' = R'$ and modify $R''$ to achieve $\sum_{j=1}^{m} r_j'' = N_{hyp}$. We want to keep the distribution of $R''$ as close as possible to the distribution of $R'$. Hence, we keep the relative distances between the individual elements in $R'$ and $R''$ similar.

If $\sum_{j=1}^{m} r_j' > N_{hyp}$, we decrement elements in $R''$ by 1 until $\sum_{j=1}^{m} r_j'' = N_{hyp}$ (Line 11). We decrement in order from lowest to highest valued element in $R'$, that is, lowest to highest weighted base model in the resulting weight vector. Thus, first trimming base models with only one repetition. Finally, if $\sum_{j=1}^{m} r_j' - N_{hyp}$ is large enough, we decrement the most repeated elements. Note, due to rounding, we must decrement each element once in the worst case. If $\sum_{j=1}^{m} r_j' < N_{hyp}$, we have to increase the value of elements in $R''$. To keep the relative distances similar, we equally distributed $N_{hyp} - \sum_{j=1}^{m} r_j'$ increments between all non-zero elements in $R''$ (Line 13). Finally, the $R''$ is transformed into a weight vector with Equation 4.

---

**Algorithm 1** The Procedure for CMA-ES-ExplicitGES

---

**Input**: Weight vector $W'$ of length $m$, the number of hypothetical iterations $N_{hyp}$
**Output**: Weight vector $W$

1: $W \leftarrow W^s$ computed with Equation 3 using $W'$      ▷ Apply softmax.
2: **for** $i = 1$ **to** $m$ **do**      ▷ Trim base models.
3:      **if** $w_i \leq \frac{0.5}{N_{hyp}}$ **then**
4:          $w_i \leftarrow 0$
5: **if** $\sum_{i=1}^{m} w_i = 0$ **then**      ▷ Fallback to unweighted average.
6:      **return** $(\frac{1}{m}, ..., \frac{1}{m})$
7: $R' \leftarrow [0 \cdots 0]$      ▷ Initialize an empty vector of repetitions.
8: **for** $i = 1$ **to** $m$ **do**      ▷ Round to nearest integer.
9:      $r_i' \leftarrow \lfloor w_i^s * N_{hyp} \rceil$
10: $R'' \leftarrow R'$
11: **if** $\sum_{j=1}^{m} r_j' > N_{hyp}$ **then**
12:      $R'' \leftarrow$ Decrement elements from lowest to highest valued element in $R''$ by 1 until $\sum_{j=1}^{m} r_j'' = N_{hyp}$
13: **if** $\sum_{j=1}^{m} r_j' < N_{hyp}$ **then**
14:      $R'' \leftarrow$ Equally distributed $N_{hyp} - \sum_{j=1}^{m} r_j'$ increments between all non-zero elements in $R''$
15: **return** $W$ computed with Equation 4 using $R''$.

---

## 4.4 Comparing Normalization Methods

We use CMA-ES-ExplicitGES for the final evaluation below because it is the only approach that is in line with GES's concepts. Nevertheless, here, we provide an additional comparison of the three

normalization methods on the same data as used in Section 3.1. We run CMA-ES, as described above, with the three different methods for normalization on the data from AutoGluon for ROC AUC. We ignore the threshold-dependent balanced accuracy because CMA-ES is not affected by overfitting for balanced accuracy. Besides normalization, the main difference to the application from Section 3 is that we do not apply softmax after aggregation anymore when we apply normalization.

First, a note regarding sparseness. On average, across all datasets for ROC AUC, ~13.2 base models exist, see Appendix F for each dataset's number. For comparison, we computed the average number of non-zero weighted base models for the ensemble methods, see Appendix H. This shows that CMA-ES without normalization has an average ensemble size, that is, the number of non-zero weighted base models, of ~12.9. In contrast, CMA-ES-ExplicitGES has an average ensemble size of ~6.3, CMA-ES-ImplicitGES of ~5.4. For context, GES has an average ensemble size of ~5.8 Hence, we conclude that CMA-ES produces dense weight vectors. While our normalization approaches are able to produce sparse vectors like GES.

Next, we repeat the statistical test performed in Section 3.1 for all normalization methods, CMA-ES, and the SingleBest, see Figure 4 in the Appendix H. We observe that all normalization methods outperform CMA-ES and that CMA-ES-ExplicitGES ranks highest. Furthermore, the different normalization methods are not statistically significantly different from each other. Only CMA-ES-ExplicitGES is significantly different from CMA-ES for multi-class.

## 5  Overall Experiments

In our final evaluation, we mirror the experiments from Section 3.1 and compare the SingleBest, GES, CMA-ES, and CMA-ES with normalization (CMA-ES-ExplicitGES). We additionally include stacking in our comparison because it is part of Auto-Sklearn's insight and used by H2O AutoML. For our implementation of stacking (Wolpert, 1992), we use a default Logistic Regression classifier from scikit-learn (Pedregosa et al., 2011) as a stacking model. We adjusted the code such that we terminate after $m * 50$ evaluations to make the method comparable to GES and CMA-ES. For CMA-ES we stick to the implementation and default hyperparameters as described in Section 3.

Besides the statistical tests, we also inspect the difference in the distributions of relative performance. Therefore, we follow the AutoML benchmark (Gijsbers et al., 2022) and use *normalized improvement* to make the scores of methods comparable across different datasets. We scale the scores for a dataset such that $-1$ is equal to the score of a baseline, here the SingleBest, and 0 is equal to the score of the best method on the dataset. We employ a variant of normalized improvement as we ran into an edge case where the normalized improvement is undefined if the difference between the single best model and the best method is 0. In our variant, for this edge case, we set everything as good as the SingleBest to $-1$ and penalize all methods worse than the baseline with $-10$; following a penalization approach like PAR10 from Algorithm Selection (Lindauer et al., 2019). We provide a formalized definition of normalized improvement in Appendix I.2.

## 6  Overall Results

Figure 2 shows the results of the statistical tests and mean rankings for the compared methods. The distribution of the relative performance is shown in Figure 3. Additionally, the performance per dataset is provided in Appendix J.

**Overall Predictive Performance**. All post hoc ensembling methods always outperform the SingleBest on average, although not always statistically significant – see Figure 2. Yet, post hoc ensembling can overfit and become worse for specific datasets, as indicated by the black dots left of the red bar and the number of outliers in square brackets in Figure 3.

*For balanced accuracy*, we observe that CMA-ES significantly beats all methods. Likewise, we observe that stacking and CMA-ES-ExplicitGES outperform GES by a small non-significant margin.

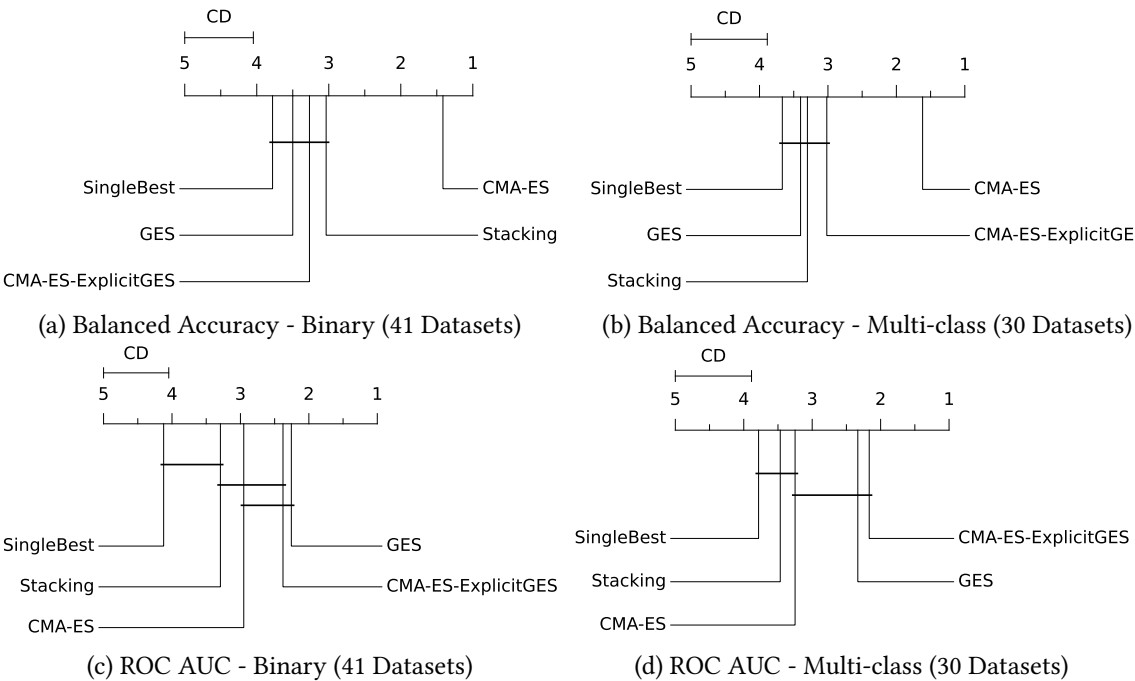

(a) Balanced Accuracy - Binary (41 Datasets)

(b) Balanced Accuracy - Multi-class (30 Datasets)

(c) ROC AUC - Binary (41 Datasets)

(d) ROC AUC - Multi-class (30 Datasets)

Figure 2: **CD Plots for all Methods**: Methods connected by a bar are not significantly different.

*For ROC AUC*, we see that GES and CMA-ES-ExplicitGES outperform all other methods and differ only by a small non-significant margin. Both are also significantly different from the SingleBest; unlike stacking. Moreover, Figure 3 shows us that CMA-ES-ExplicitGES has similar or better relative performance distributions than GES (see the medians and whiskers).

**Normalization to Combat Overfitting**. See Table 2 to inspect overfitting for CMA-ES-ExplictGES. See Appendix G.2 for an overview of the rank change for all compared methods. In general, CMA-ES-ExplictGES's mean rank, compared to GES and the SingleBest, changes only minimally between validation and test data. Showing us that it overfits less than CMA-ES (compare to Table 1, Section 3.2). As before, the SingleBest is always the worst-ranked method. GES is worse than CMA-ES-ExplictGES on test data for all but ROC AUC Binary. On validation data, however, GES is better than CMA-ES-ExplictGES in all cases except for ROC AUC multi-class, where it is tied. Now, GES is *more affected by overfitting* than CMA-ES with normalization.

**No Free Lunch**. CMA-ES-ExplictGES for balanced accuracy ranks worse than CMA-ES but better than GES. In contrast, CMA-ES-ExplictGES ranks better than CMA-ES for ROC AUC. A decrease in performance for balanced accuracy was to be expected as the normalization method constrained the solutions of CMA-ES to be sparse and pseudo-discrete to combat overfitting, but CMA-ES did not overfit for balanced accuracy. Moreover, it indicates that satisfying these properties of GES for balanced accuracy is suboptimal. Hence, our results also indicate the need to select the best method per task and metric instead of always using the same method; in line with the *no free lunch theorem*. Likewise, the drastic differences in performance of the methods between metrics suggest that the optimization landscapes, and the impact of overfitting on them, differ drastically.

## 7 Conclusion

Greedy ensemble selection (GES) (Caruana et al., 2004) is often used for post hoc ensembling in AutoML; likely as a result of Auto-Sklearn 1's (Feurer et al., 2015) reported insight that GES is superior to potential alternatives, like gradient-free numerical optimization, for post hoc ensembling.

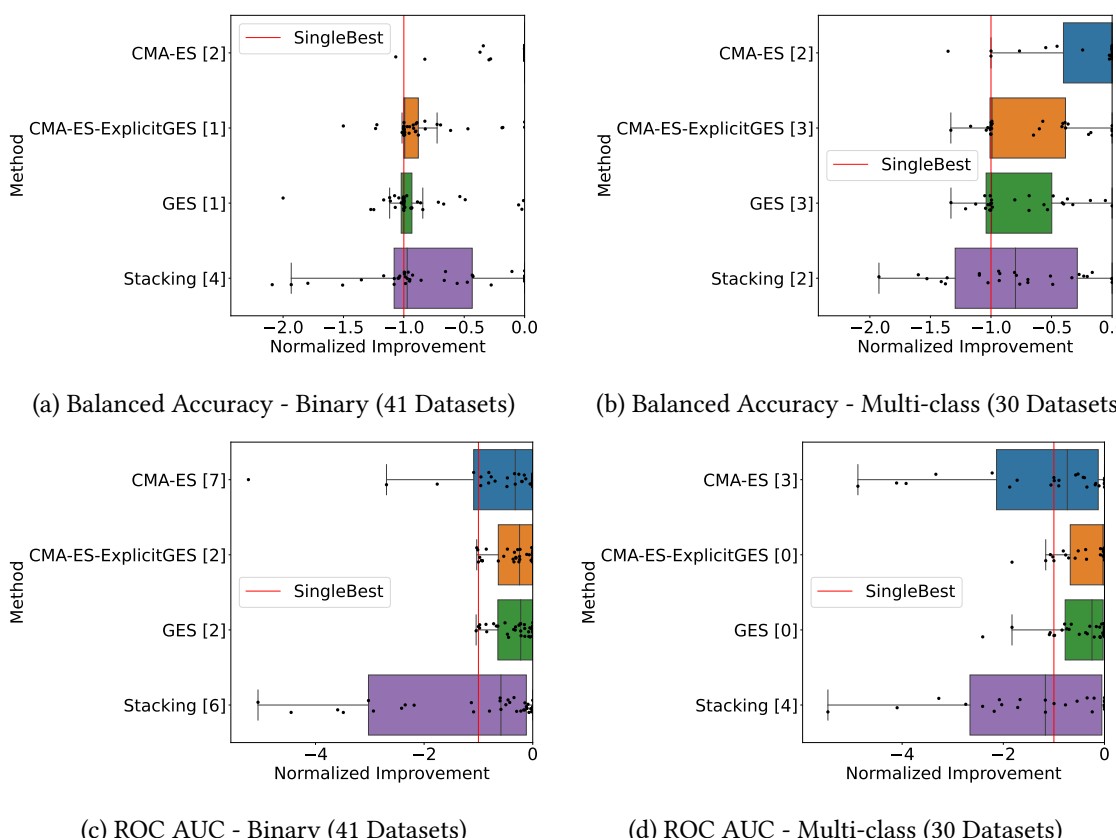

(a) Balanced Accuracy - Binary (41 Datasets)

(b) Balanced Accuracy - Multi-class (30 Datasets)

(c) ROC AUC - Binary (41 Datasets)

(d) ROC AUC - Multi-class (30 Datasets)

Figure 3: **Normalized Improvement Boxplots**: Higher normalized improvement is better. Each black point represents the improvement for one dataset. A value smaller than −1 is worse than the single best model (red vertical line), while 0 is the best observed value. The number in square brackets counts the outliers of a method left of the plot's boundary.

Table 2: Mean rank change for CMA-ES-ExplictGES compared to GES and SingleBest. In the case of a tie for the absolute rank, we assign all tied values the average of their tie-broken ranks.

| Metric | Task Type | Mean $Rank_{Validation}$ | Mean $Rank_{Test}$ | Absolute Rank (Val → Test) |
|---|---|---|---|---|
| Balanced Accuracy | Binary | 1.74 | 1.78 | 2.0 → 1.0 |
| Balanced Accuracy | Multi-class | 1.73 | 1.78 | 2.0 → 1.0 |
| ROC AUC | Binary | 1.63 | 1.70 | 2.0 → 2.0 |
| ROC AUC | Multi-class | 1.50 | 1.57 | 1.5 → 1.0 |

In this paper, we have shown that Auto-Sklearn's insight w.r.t. overfitting depends on the metric when tested for an AutoML system with higher-quality validation data than Auto-Sklearn, e.g., AutoGluon (Erickson et al., 2020). Indeed, for the metric ROC AUC, GES does not overfit meaningfully, while gradient-free numerical optimization, e.g., CMA-ES (Hansen and Auger, 2014; Hansen, 2016), overfits drastically. However, for balanced accuracy, CMA-ES does not overfit and outperforms GES.

As a direct consequence, we were motivated to find a method that combats the overfitting of CMA-ES for ROC AUC. Therefore, we proposed a novel normalization method, is inspired by GES, which successfully salvages CMA-ES for ROC AUC by making CMA-ES perform better than or similar to GES.

**Acknowledgements**. The CPU nodes of the OMNI cluster of the University of Siegen (North Rhine-Westphalia, Germany) were used for all experiments presented in this work.

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

## B  Limitations

We note that our work is limited with respect to the following points: 1) we did not explore variations (w.r.t. hyperparameters or implementation) of CMA-ES in our work; 2) we considered overfitting with respect to mean rank change between validation and test data, but did not consider other concepts of overfitting; 3) we only looked at normalization to combat overfitting for CMA-ES and were not able to compare normalization to using constraints during optimization; 4) we only provided a high-level theoretical analysis of GES and were not able to provide more fundamental work or proofs ; and 5) we only evaluated our approach for AutoGluon, one AutoML system with its specific approach to AutoML.

## C  Broader Impact Statement

After careful reflection, we determine that this work presents *almost* no notable or new negative impacts to society or the environment that are not already present for existing state-of-the-art AutoML systems. This follows from our work being mostly domain-independent, abstract, and methodical. We only proposed to replace one component of an AutoML system such that the predictive performance improves. Nevertheless, we would like to remark that our work might prompt others to use a default application of CMA-ES instead of GES for a metric like balanced accuracy. This might have a negative impact on the environment because this would likely increase the inference time and size of the final ensemble proposed by AutoML systems.

In contrast – as a trade-off – we see the positive impact that higher predictive performance with CMA-ES could better support decisions made with AutoML systems. Moreover, we believe that our work might help to understand GES, the currently most used method, better; such that its performance and behaviour becomes more explainable.

## D  Used Assets: Essential Python Frameworks for the Implementation and Experiments

The following frameworks were essential for our implementation and experiments:

- AutoGluon (Erickson et al., 2020), Version: 0.6.2, Apache-2.0 License; We used AutoGluon to generate base models for post hoc ensembling.

- pycma (Hansen et al., 2019), Version 3.2.2, BSD 3-Clause License; We used pycma for CMA-ES.

- Assembled (Purucker and Beel, 2022), Version 0.0.4, MIT License; We used Assembled to store the base models generated with AutoGluon and to run our ensemble-related experiments.

## E  DOIs for Data and Code

The following assets were newly created as part of our experiments:

- The code for our experiments: `https://doi.org/10.6084/m9.figshare.23609226`.

- The prediction data of base models collected by running AutoGluon on the classification datasets from the AutoML benchmark: `https://doi.org/10.6084/m9.figshare.23609361`.

# F Data Overview

See Table 3 for an overview of the used datasets and their characteristics. Additionally, the table shows the mean number of base models and the mean number of distinct algorithms generated by AutoGluon for the dataset for each metric (mean over the 10 folds of a dataset).

Table 3: Data Overview

| Dataset Name | OpenML Task ID | #instances | #features | #classes | Memory (GB) | Mean # Base Models | | Mean # Distinct Algorithms | |
|---|---|---|---|---|---|---|---|---|---|
| | | | | | | Balanced Accuracy | ROC AUC | Balanced Accuracy | ROC AUC |
| yeast | 2073 | 1484 | 9 | 10 | 32 | 21.0 | 21.3 | 12.0 | 12.1 |
| KDDCup09_appetency | 3945 | 50000 | 231 | 2 | 32 | 11.0 | 11.0 | 11.0 | 11.0 |
| covertype | 7593 | 581012 | 55 | 7 | 64 | 13.3 | 12.9 | 8.3 | 8.0 |
| amazon-commerce-reviews | 10090 | 1500 | 10001 | 50 | 32 | 8.3 | 8.7 | 5.3 | 5.7 |
| Australian | 146818 | 690 | 15 | 2 | 32 | 13.0 | 13.0 | 13.0 | 13.0 |
| wilt | 146820 | 4839 | 6 | 2 | 32 | 12.0 | 12.0 | 12.0 | 12.0 |
| numerai28.6 | 167120 | 96320 | 22 | 2 | 32 | 12.0 | 12.0 | 12.0 | 12.0 |
| phoneme | 168350 | 5404 | 6 | 2 | 32 | 12.0 | 11.9 | 12.0 | 11.9 |
| credit-g | 168757 | 1000 | 21 | 2 | 32 | 13.0 | 13.0 | 13.0 | 13.0 |
| steel-plates-fault | 168784 | 1941 | 28 | 7 | 32 | 21.0 | 21.0 | 12.0 | 12.0 |
| APSFailure | 168868 | 76000 | 171 | 2 | 32 | 12.0 | 12.0 | 12.0 | 12.0 |
| dilbert | 168909 | 10000 | 2001 | 5 | 32 | 12.9 | 12.7 | 7.9 | 8.3 |
| fabert | 168910 | 8237 | 801 | 7 | 32 | 19.8 | 19.8 | 11.0 | 11.0 |
| jasmine | 168911 | 2984 | 145 | 2 | 32 | 12.3 | 12.9 | 12.3 | 12.9 |
| airlines | 189354 | 539383 | 8 | 2 | 64 | 9.0 | 8.9 | 9.0 | 8.9 |
| dionis | 189355 | 416188 | 61 | 355 | 128 | 4.0 | 4.2 | 3.0 | 3.6 |
| albert | 189356 | 425240 | 79 | 2 | 64 | 7.0 | 7.0 | 7.0 | 7.0 |
| gina | 189922 | 3153 | 971 | 2 | 32 | 12.0 | 12.0 | 12.0 | 12.0 |
| ozone-level-8hr | 190137 | 2534 | 73 | 2 | 32 | 13.0 | 13.0 | 13.0 | 13.0 |
| vehicle | 190146 | 846 | 19 | 4 | 32 | 24.0 | 24.0 | 13.0 | 13.0 |
| madeline | 190392 | 3140 | 260 | 2 | 32 | 12.0 | 12.3 | 12.0 | 12.3 |
| philippine | 190410 | 5832 | 309 | 2 | 32 | 12.0 | 12.0 | 12.0 | 12.0 |
| ada | 190411 | 4147 | 49 | 2 | 32 | 12.0 | 12.0 | 12.0 | 12.0 |
| arcene | 190412 | 100 | 10001 | 2 | 32 | 13.0 | 13.0 | 13.0 | 13.0 |
| jannis | 211979 | 83733 | 55 | 4 | 32 | 14.9 | 15.3 | 9.2 | 9.3 |
| Diabetes130US | 211986 | 101766 | 50 | 3 | 32 | 17.9 | 18.1 | 10.8 | 10.9 |
| micro-mass | 359953 | 571 | 1301 | 20 | 32 | 13.0 | 13.0 | 13.0 | 13.0 |
| eucalyptus | 359954 | 736 | 20 | 5 | 32 | 13.0 | 13.0 | 13.0 | 13.0 |
| blood-transfusion-service-center | 359955 | 748 | 5 | 2 | 32 | 13.0 | 13.0 | 13.0 | 13.0 |
| qsar-biodeg | 359956 | 1055 | 42 | 2 | 32 | 13.0 | 13.0 | 13.0 | 13.0 |
| cnae-9 | 359957 | 1080 | 857 | 9 | 32 | 20.0 | 20.0 | 11.0 | 11.0 |
| pc4 | 359958 | 1458 | 38 | 2 | 32 | 13.0 | 13.0 | 13.0 | 13.0 |
| cmc | 359959 | 1473 | 10 | 3 | 32 | 21.8 | 21.6 | 12.0 | 12.0 |
| car | 359960 | 1728 | 7 | 4 | 32 | 18.4 | 18.0 | 9.2 | 9.0 |
| mfeat-factors | 359961 | 2000 | 217 | 10 | 32 | 20.0 | 20.0 | 11.0 | 11.0 |
| kc1 | 359962 | 2109 | 22 | 2 | 32 | 12.8 | 13.0 | 12.8 | 13.0 |
| segment | 359963 | 2310 | 20 | 7 | 32 | 21.0 | 21.0 | 12.0 | 12.0 |
| dna | 359964 | 3186 | 181 | 3 | 32 | 19.0 | 19.0 | 10.0 | 10.0 |
| kr-vs-kp | 359965 | 3196 | 37 | 2 | 32 | 10.0 | 10.0 | 10.0 | 10.0 |
| Internet-Advertisements | 359966 | 3279 | 1559 | 2 | 32 | 12.0 | 12.0 | 12.0 | 12.0 |
| Bioresponse | 359967 | 3751 | 1777 | 2 | 32 | 12.0 | 12.0 | 12.0 | 12.0 |
| churn | 359968 | 5000 | 21 | 2 | 32 | 12.0 | 12.0 | 12.0 | 12.0 |
| first-order-theorem-proving | 359969 | 6118 | 52 | 6 | 32 | 20.1 | 20.1 | 11.1 | 11.1 |
| GesturePhaseSegmentationProcessed | 359970 | 9873 | 33 | 5 | 32 | 20.0 | 20.4 | 11.0 | 11.2 |
| PhishingWebsites | 359971 | 11055 | 31 | 2 | 32 | 10.0 | 10.0 | 10.0 | 10.0 |
| sylvine | 359972 | 5124 | 21 | 2 | 32 | 12.0 | 12.0 | 12.0 | 12.0 |
| christine | 359973 | 5418 | 1637 | 2 | 32 | 12.0 | 12.0 | 12.0 | 12.0 |
| wine-quality-white | 359974 | 4898 | 12 | 7 | 32 | 21.0 | 21.0 | 12.0 | 12.0 |
| Satellite | 359975 | 5100 | 37 | 2 | 32 | 12.0 | 12.0 | 12.0 | 12.0 |
| Fashion-MNIST | 359976 | 70000 | 785 | 10 | 64 | 12.1 | 13.0 | 8.5 | 8.2 |
| connect-4 | 359977 | 67557 | 43 | 3 | 32 | 16.2 | 16.3 | 9.0 | 9.0 |
| Amazon_employee_access | 359979 | 32769 | 10 | 2 | 32 | 9.1 | 10.0 | 9.1 | 10.0 |
| nomao | 359980 | 34465 | 119 | 2 | 32 | 12.0 | 10.0 | 12.0 | 10.0 |
| jungle_chess_2pcs_raw_endgame_complete | 359981 | 44819 | 7 | 3 | 32 | 19.5 | 19.9 | 11.0 | 11.0 |
| bank-marketing | 359982 | 45211 | 17 | 2 | 32 | 12.0 | 12.0 | 12.0 | 12.0 |
| adult | 359983 | 48842 | 15 | 2 | 32 | 12.0 | 11.9 | 12.0 | 11.9 |
| helena | 359984 | 65196 | 28 | 100 | 32 | 7.7 | 7.9 | 5.0 | 5.0 |
| volkert | 359985 | 58310 | 181 | 10 | 32 | 13.9 | 12.5 | 8.9 | 8.6 |
| robert | 359986 | 10000 | 7201 | 10 | 64 | 9.7 | 9.3 | 7.6 | 7.3 |
| shuttle | 359987 | 58000 | 10 | 7 | 32 | 18.9 | 19.0 | 11.0 | 11.0 |
| guillermo | 359988 | 20000 | 4297 | 2 | 32 | 9.0 | 9.0 | 9.0 | 9.0 |
| riccardo | 359989 | 20000 | 4297 | 2 | 32 | 10.1 | 9.0 | 10.1 | 9.0 |
| MiniBooNE | 359990 | 130064 | 51 | 2 | 32 | 10.3 | 10.2 | 10.3 | 10.2 |
| kick | 359991 | 72983 | 33 | 2 | 32 | 11.7 | 12.0 | 11.7 | 12.0 |
| Click_prediction_small | 359992 | 39948 | 12 | 2 | 32 | 11.8 | 12.4 | 11.8 | 12.4 |
| okcupid-stem | 359993 | 50789 | 20 | 3 | 32 | 18.6 | 19.8 | 11.0 | 11.4 |
| sf-police-incidents | 359994 | 2215023 | 9 | 2 | 64 | 11.0 | 7.2 | 11.0 | 7.2 |
| KDDCup99 | 360112 | 4898431 | 42 | 23 | 128 | 10.3 | 7.6 | 8.5 | 6.9 |
| porto-seguro | 360113 | 595212 | 58 | 2 | 64 | 6.2 | 2.2 | 6.2 | 2.2 |
| Higgs | 360114 | 1000000 | 29 | 2 | 64 | 3.0 | 3.0 | 3.0 | 3.0 |
| KDDCup09-Upselling | 360975 | 50000 | 14892 | 2 | 128 | 10.5 | 9.0 | 10.5 | 9.0 |

# G Overview of Rank Change from Validation to Test Data

This section provides an overview of the rank change from validation to test data for the compared methods to inspect overfitting. Table 4 gives the overview for the comparison made in Section 3.2. Table 5 gives the overview for the comparison made in Section 6.

## G.1 Supplements for Section 3.2

Table 4: Mean rank change from validation to test data for CMA-ES, GES, and SingleBest. We denote the mean rank on validation data with *MRV* and the mean rank on test data with *MRT*.

| Method | MRV | MRT | Absolute Rank (Val → Test) |
|---|---|---|---|
| CMA-ES | 1.00 | 1.12 | 1.00 ->1.00 |
| GES | 2.06 | 2.44 | 2.00 ->2.50 |
| SingleBest | 2.94 | 2.44 | 3.00 ->2.50 |

(a) Balanced Accuracy - Binary

| Method | MRV | MRT | Absolute Rank (Val → Test) |
|---|---|---|---|
| CMA-ES | 1.03 | 1.25 | 1.00 ->1.00 |
| GES | 1.97 | 2.33 | 2.00 ->2.00 |
| SingleBest | 3.00 | 2.42 | 3.00 ->3.00 |

(b) Balanced Accuracy - Multi-class

| Method | MRV | MRT | Absolute Rank (Val → Test) |
|---|---|---|---|
| CMA-ES | 1.02 | 1.83 | 1.00 ->2.00 |
| GES | 1.99 | 1.52 | 2.00 ->1.00 |
| SingleBest | 2.99 | 2.65 | 3.00 ->3.00 |

(c) ROC AUC - Binary

| Method | MRV | MRT | Absolute Rank (Val → Test) |
|---|---|---|---|
| CMA-ES | 1.42 | 2.12 | 1.00 ->2.00 |
| GES | 1.60 | 1.40 | 2.00 ->1.00 |
| SingleBest | 2.98 | 2.48 | 3.00 ->3.00 |

(d) ROC AUC - Multi-class

## G.2 Supplements for Section 6

Table 5: Mean rank change from validation to test data for CMA-ES-ExplicitGES, GES, and SingleBest. We denote the mean rank on validation data with *MRV* and the mean rank on test data with *MRT*. In the case of a tie for the absolute rank, we assign all tied values the average of their randomly tie-broken ranks. As a result, a rank of 1.5 for validation data for ROC AUC multi-class occurs since CMA-ES-ExplicitGES and GES are tied.

| Method | MRV | MRT | Absolute Rank (Val → Test) |
|---|---|---|---|
| CMA-ES-ExplicitGES | 1.74 | 1.78 | 2.00 ->1.00 |
| GES | 1.40 | 2.00 | 1.00 ->2.00 |
| SingleBest | 2.85 | 2.22 | 3.00 ->3.00 |

(a) Balanced Accuracy - Binary

| Method | MRV | MRT | Absolute Rank (Val → Test) |
|---|---|---|---|
| CMA-ES-ExplicitGES | 1.73 | 1.78 | 2.00 ->1.00 |
| GES | 1.27 | 2.07 | 1.00 ->2.00 |
| SingleBest | 3.00 | 2.15 | 3.00 ->3.00 |

(b) Balanced Accuracy - Multi-class

| Method | MRV | MRT | Absolute Rank (Val → Test) |
|---|---|---|---|
| CMA-ES-ExplicitGES | 1.63 | 1.70 | 2.00 ->2.00 |
| GES | 1.39 | 1.50 | 1.00 ->1.00 |
| SingleBest | 2.98 | 2.80 | 3.00 ->3.00 |

(c) ROC AUC - Binary

| Method | MRV | MRT | Absolute Rank (Val → Test) |
|---|---|---|---|
| CMA-ES-ExplicitGES | 1.50 | 1.57 | 1.50 ->1.00 |
| GES | 1.50 | 1.73 | 1.50 ->2.00 |
| SingleBest | 3.00 | 2.70 | 3.00 ->3.00 |

(d) ROC AUC - Multi-class

# H Comparison of Normalization Methods

See Figure 4 for a comparison of the three proposed normalization methods following the experiments described in Section 3.1.

The difference between the presented methods shows a small ablation study of our approaches w.r.t. satisfying the properties of GES, *pseudo-discrete* and *sparse* (specified in Section 4.1). CMA-ES and CMA-ES-Softmax are versions without either property; CMA-ES-ImplicitGES satisfies only sparseness; and CMA-ES-ExplicitGES satisfies both properties. Only the method that satisfies both

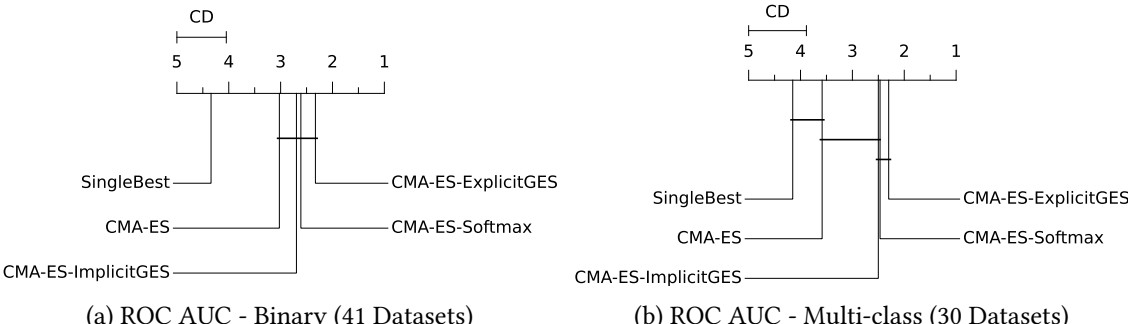

(a) ROC AUC - Binary (41 Datasets)        (b) ROC AUC - Multi-class (30 Datasets)

Figure 4: **CD Plots Comparing the Normalization Methods for ROC AUC**: Mean rank of the methods (lower is better). Methods connected by a bar are not significantly different.

Table 6: The average ensemble size (non-zero weighted base models) for CMA-ES, CMA-ES-ExplicitGES, CMA-ES-ImplicitGES, and GES for binary and multi-class classification with ROC AUC.

| Task Type | CMA-ES | CMA-ES-ExplicitGES | CMA-ES-ImplicitGES | GES |
|---|---|---|---|---|
| Binary Classification | 10.973 | 5.263 | 4.471 | 4.761 |
| Multi-class Classification | 14.827 | 7.300 | 6.310 | 6.850 |
| Average | ~12.9 | ~6.3 | ~5.4 | ~5.8 |

properties, CMA-ES-ExplicitGES, is significantly different from CMA-ES for multi-class and always has the best mean rank.

To analyze the effect of trimming base models on the size of the ensemble, we show the average ensemble size in Table 6.

# I Supplements for Experiments Following the AutoML Benchmark (Gijsbers et al., 2022)

## I.1 Statistical Test with Critical Difference Plots

Following the AutoML benchmark (Gijsbers et al., 2022), we perform a statistical test using a Friedman test with a Nemenyi post hoc test ($\alpha = 0.05$). We implemented the tests re-using code from Autorank (Herbold, 2020).

We first calculate the mean rank of each method for each collection of datasets, i.e., the subset of datasets for binary or multi-class classification for both metrics. Then, we use the Friedman test as an omnibus test to try to reject the null hypothesis that there is no difference between the methods. Only if the Friedman test is significant and rejects the null hypothesis, we perform a Nemenyi post hoc test. The test calculates a critical difference (CD). Finally, we determine if the difference between methods is significant by verifying that their difference in mean rank is greater than the CD. Otherwise, the difference is not significant. We show the results of the Nemenyi post hoc test using CD plots, whereby a horizontal bar connects methods that are not significantly different.

## I.2 Normalized Improvement

Our implementation of normalized improvement follows the AutoML benchmark (Gijsbers et al., 2022). That is, we scale the scores for a dataset such that $-1$ is equal to the score of the single best model, and 0 is equal to the score of the best method on the dataset.

Formally, we normalise the score $s_D$ of a method for a dataset $D$ using:

$$\frac{s_D - s_D^b}{s_D^* - s_D^b} - 1, \tag{5}$$

with the score of the baseline $s_D^b$ and the best-observed score for the dataset $s_D^*$. We assume that higher scores are always better.

We extend this definition for the edge cases where no method is better than the baseline, *i.e.*, $s_D^* - s_D^b = 0$. We suppose that this edge case never happened in the AutoML benchmark. Otherwise, their definition and implementation would have crashed. In our setting, such an edge case can happen due to overfitting such that the ensemble methods becomes worse than the single best model.

If the edge case happens, we set the score of all methods worse than the baseline to $-10$, following a penalization-like approach (*e.g.*, PAR10 from Algorithm Selection (Lindauer et al., 2019)). Methods for which $s_D - s_D^b = 0$ holds are assigned a score of $-1$.

## J Overview of Performance per Dataset

Here we provide the mean and standard deviation over all folds per dataset. The different combinations of metric and classification tasks are split into separate tables, see Tables 7,8, 9, and 10.

Table 7: **Balanced Accuracy - Binary**: The mean and standard deviation of the test score over all folds for each method. The best method per dataset is shown in bold.

| Dataset | CMA-ES | CMA-ES-ExplicitGES | GES | SingleBest | Stacking |
|---|---|---|---|---|---|
| APSFailure | **0.9563 (± 0.0118)** | 0.8957 (± 0.0207) | 0.8957 (± 0.0205) | 0.8961 (± 0.0208) | 0.8977 (± 0.0236) |
| Amazon_employee_access | **0.8262 (± 0.0153)** | 0.6883 (± 0.0104) | 0.6884 (± 0.0105) | 0.6883 (± 0.0104) | 0.6902 (± 0.0147) |
| Australian | 0.8598 (± 0.0293) | 0.8615 (± 0.0237) | 0.8579 (± 0.0357) | 0.8605 (± 0.0174) | **0.871 (± 0.0247)** |
| Bioresponse | **0.8085 (± 0.0175)** | 0.805 (± 0.0218) | 0.8052 (± 0.0228) | 0.8045 (± 0.0222) | 0.8066 (± 0.02) |
| Click_prediction_small | **0.6461 (± 0.0081)** | 0.5535 (± 0.0052) | 0.5535 (± 0.0052) | 0.5535 (± 0.0052) | 0.5504 (± 0.0057) |
| Higgs | 0.7414 (± 0.0012) | 0.7408 (± 0.001) | 0.7408 (± 0.001) | 0.7407 (± 0.001) | **0.7417 (± 0.0012)** |
| Internet-Advertisements | **0.9454 (± 0.0251)** | 0.9438 (± 0.0247) | 0.9452 (± 0.0247) | 0.9365 (± 0.0243) | 0.9445 (± 0.0219) |
| KDDCup09-Upselling | **0.7906 (± 0.0076)** | 0.7818 (± 0.0103) | 0.7823 (± 0.0102) | 0.7817 (± 0.01) | 0.7683 (± 0.009) |
| KDDCup09_appetency | **0.6622 (± 0.0622)** | 0.5 (± 0.0) | 0.5 (± 0.0) | 0.5 (± 0.0) | 0.5005 (± 0.0018) |
| MiniBooNE | **0.9442 (± 0.0051)** | 0.9399 (± 0.0028) | 0.94 (± 0.0028) | 0.9399 (± 0.0028) | 0.9364 (± 0.0031) |
| PhishingWebsites | **0.9723 (± 0.0037)** | 0.972 (± 0.0046) | 0.9722 (± 0.0049) | 0.9705 (± 0.0046) | 0.9715 (± 0.0045) |
| Satellite | **0.9281 (± 0.0418)** | 0.8317 (± 0.0704) | 0.8248 (± 0.0712) | 0.8316 (± 0.0703) | 0.832 (± 0.078) |
| ada | **0.8102 (± 0.0299)** | 0.7956 (± 0.0293) | 0.7953 (± 0.0289) | 0.7943 (± 0.0277) | 0.7887 (± 0.0282) |
| adult | **0.842 (± 0.0069)** | 0.8002 (± 0.0069) | 0.7993 (± 0.0069) | 0.8002 (± 0.0069) | 0.8007 (± 0.0067) |
| airlines | 0.6422 (± 0.0025) | **0.6573 (± 0.0016)** | 0.6573 (± 0.0017) | 0.6569 (± 0.0014) | 0.649 (± 0.0017) |
| albert | 0.7049 (± 0.0021) | 0.7046 (± 0.0023) | 0.7047 (± 0.0023) | 0.7043 (± 0.0026) | **0.7051 (± 0.0024)** |
| arcene | **0.8383 (± 0.1616)** | 0.7967 (± 0.1904) | 0.8075 (± 0.1911) | 0.7808 (± 0.1646) | 0.785 (± 0.1807) |
| bank-marketing | **0.8722 (± 0.0097)** | 0.7447 (± 0.0117) | 0.7448 (± 0.012) | 0.7448 (± 0.0116) | 0.7348 (± 0.0139) |
| blood-transfusion-service-center | 0.6315 (± 0.0378) | 0.642 (± 0.0476) | 0.6419 (± 0.0437) | **0.6514 (± 0.0398)** | 0.6233 (± 0.0403) |
| christine | **0.7556 (± 0.0133)** | 0.7521 (± 0.0173) | 0.7517 (± 0.0163) | 0.7517 (± 0.0163) | 0.7523 (± 0.0191) |
| churn | **0.9075 (± 0.0248)** | 0.8912 (± 0.0211) | 0.8896 (± 0.0188) | 0.8809 (± 0.0248) | 0.9001 (± 0.024) |
| credit-g | **0.689 (± 0.0369)** | 0.6843 (± 0.0494) | 0.6855 (± 0.0474) | 0.684 (± 0.0472) | 0.6843 (± 0.0432) |
| gina | **0.9607 (± 0.0153)** | 0.9563 (± 0.0179) | 0.9563 (± 0.0179) | 0.9563 (± 0.0179) | 0.956 (± 0.0187) |
| guillermo | **0.8411 (± 0.0102)** | 0.8171 (± 0.0094) | 0.8171 (± 0.0094) | 0.8171 (± 0.0094) | 0.8307 (± 0.0111) |
| jasmine | 0.8167 (± 0.0169) | 0.8167 (± 0.0186) | 0.8154 (± 0.0183) | 0.8154 (± 0.0183) | **0.8231 (± 0.0182)** |
| kc1 | **0.7021 (± 0.0329)** | 0.6423 (± 0.0357) | 0.6423 (± 0.0357) | 0.6423 (± 0.0357) | 0.6453 (± 0.0415) |
| kick | **0.6985 (± 0.0139)** | 0.6231 (± 0.0068) | 0.6231 (± 0.0068) | 0.6231 (± 0.0068) | 0.6258 (± 0.0086) |
| kr-vs-kp | 0.9947 (± 0.005) | 0.9934 (± 0.0045) | 0.9935 (± 0.0042) | 0.9932 (± 0.005) | **0.9956 (± 0.0051)** |
| madeline | **0.8767 (± 0.011)** | 0.8694 (± 0.0192) | 0.8688 (± 0.0185) | 0.8691 (± 0.0187) | 0.8716 (± 0.0176) |
| nomao | **0.9733 (± 0.0031)** | 0.9668 (± 0.0027) | 0.9666 (± 0.0027) | 0.9666 (± 0.0027) | 0.9663 (± 0.0032) |
| numerai28.6 | **0.5209 (± 0.0047)** | 0.5192 (± 0.0046) | 0.5194 (± 0.0045) | 0.5196 (± 0.0042) | 0.5202 (± 0.0046) |
| ozone-level-8hr | **0.7963 (± 0.0398)** | 0.7203 (± 0.0447) | 0.7203 (± 0.0447) | 0.7201 (± 0.045) | 0.6814 (± 0.0649) |
| pc4 | **0.8571 (± 0.0412)** | 0.8011 (± 0.0561) | 0.8014 (± 0.0551) | 0.8019 (± 0.0526) | 0.7417 (± 0.0598) |
| philippine | 0.7819 (± 0.019) | 0.7814 (± 0.0149) | 0.7795 (± 0.0161) | 0.7798 (± 0.016) | **0.7827 (± 0.0167)** |
| phoneme | **0.9141 (± 0.0148)** | 0.8931 (± 0.0191) | 0.8938 (± 0.0178) | 0.8905 (± 0.0216) | 0.8987 (± 0.0206) |
| porto-seguro | **0.5487 (± 0.0425)** | 0.5001 (± 0.0003) | 0.5 (± 0.0002) | 0.5001 (± 0.0003) | 0.5004 (± 0.0006) |
| qsar-biodeg | 0.8568 (± 0.0335) | 0.8504 (± 0.034) | 0.8469 (± 0.0331) | 0.849 (± 0.0342) | **0.8568 (± 0.0379)** |
| riccardo | **0.9986 (± 0.0008)** | 0.9986 (± 0.0004) | 0.9985 (± 0.0006) | 0.9985 (± 0.0007) | 0.9984 (± 0.0007) |
| sf-police-incidents | **0.6279 (± 0.0117)** | 0.5255 (± 0.001) | 0.5256 (± 0.0009) | 0.5248 (± 0.0009) | 0.5207 (± 0.0005) |
| sylvine | **0.9518 (± 0.0073)** | 0.95 (± 0.0049) | 0.9504 (± 0.005) | 0.9506 (± 0.0051) | 0.9504 (± 0.0068) |
| wilt | **0.9542 (± 0.0312)** | 0.9232 (± 0.0482) | 0.9272 (± 0.0401) | 0.9291 (± 0.0382) | 0.9057 (± 0.0479) |

Table 8: **Balanced Accuracy - Multi-class**: The mean and standard deviation of the test score over all folds for each method. The best method per dataset is shown in bold.

| Dataset | CMA-ES | CMA-ES-ExplicitGES | GES | SingleBest | Stacking |
|---|---|---|---|---|---|
| Diabetes130US | **0.4946 (± 0.0066)** | 0.4498 (± 0.0032) | 0.4495 (± 0.0038) | 0.4493 (± 0.0037) | 0.4526 (± 0.0049) |
| Fashion-MNIST | **0.9112 (± 0.0037)** | 0.9098 (± 0.0034) | 0.91 (± 0.0033) | 0.909 (± 0.0036) | 0.9097 (± 0.0044) |
| GesturePhaseSegmentationProcessed | **0.737 (± 0.0202)** | 0.7146 (± 0.0182) | 0.7141 (± 0.0182) | 0.7152 (± 0.0171) | 0.7204 (± 0.0154) |
| KDDCup99 | 0.7672 (± 0.0356) | **0.7681 (± 0.0358)** | 0.7607 (± 0.0392) | 0.7268 (± 0.0421) | 0.7297 (± 0.0387) |
| amazon-commerce-reviews | **0.8547 (± 0.0301)** | 0.85 (± 0.028) | 0.8533 (± 0.029) | 0.8307 (± 0.0352) | 0.8353 (± 0.0314) |
| car | 0.997 (± 0.0057) | 0.9954 (± 0.0112) | 0.9949 (± 0.0115) | 0.9954 (± 0.0112) | **0.9989 (± 0.0023)** |
| cmc | 0.5377 (± 0.0453) | **0.538 (± 0.0448)** | 0.5298 (± 0.0441) | 0.526 (± 0.0375) | 0.5321 (± 0.039) |
| cnae-9 | 0.9611 (± 0.0246) | 0.962 (± 0.0245) | **0.9657 (± 0.0175)** | 0.9556 (± 0.0275) | 0.963 (± 0.0214) |
| connect-4 | **0.7834 (± 0.0074)** | 0.71 (± 0.0081) | 0.7096 (± 0.0081) | 0.7101 (± 0.0078) | 0.7082 (± 0.0058) |
| covertype | **0.9707 (± 0.0032)** | 0.9579 (± 0.0034) | 0.9577 (± 0.0035) | 0.9579 (± 0.0037) | 0.953 (± 0.0029) |
| dilbert | **0.9944 (± 0.0012)** | 0.9934 (± 0.0015) | 0.9936 (± 0.0027) | 0.992 (± 0.0028) | 0.994 (± 0.002) |
| dionis | **0.8354 (± 0.0033)** | 0.8315 (± 0.0022) | 0.8316 (± 0.0021) | 0.826 (± 0.0018) | 0.8332 (± 0.0015) |
| dna | **0.9661 (± 0.0117)** | 0.9653 (± 0.011) | 0.9635 (± 0.0096) | 0.9612 (± 0.0085) | 0.9645 (± 0.0141) |
| eucalyptus | 0.691 (± 0.0472) | 0.6806 (± 0.0365) | 0.6799 (± 0.0418) | 0.6901 (± 0.0476) | **0.6937 (± 0.0673)** |
| fabert | **0.7164 (± 0.0094)** | 0.7094 (± 0.0108) | 0.7102 (± 0.0093) | 0.7104 (± 0.0124) | 0.7123 (± 0.0124) |
| first-order-theorem-proving | **0.5121 (± 0.0132)** | 0.497 (± 0.0282) | 0.4941 (± 0.0205) | 0.4858 (± 0.0233) | 0.4913 (± 0.0229) |
| helena | **0.2491 (± 0.0057)** | 0.2411 (± 0.0066) | 0.2414 (± 0.0069) | 0.2283 (± 0.0057) | 0.2092 (± 0.004) |
| jannis | **0.6546 (± 0.0129)** | 0.5691 (± 0.0046) | 0.5691 (± 0.0046) | 0.5692 (± 0.0051) | 0.5746 (± 0.0048) |
| jungle_chess_2pcs_raw_endgame_complete | **0.9808 (± 0.0047)** | 0.9703 (± 0.0065) | 0.9707 (± 0.0065) | 0.9707 (± 0.0063) | 0.9757 (± 0.0053) |
| mfeat-factors | 0.9805 (± 0.006) | 0.98 (± 0.0062) | 0.9815 (± 0.0047) | **0.9825 (± 0.0054)** | 0.979 (± 0.0088) |
| micro-mass | 0.9189 (± 0.0403) | **0.9227 (± 0.0378)** | 0.9152 (± 0.0436) | 0.9072 (± 0.0517) | 0.9017 (± 0.0401) |
| okcupid-stem | **0.7 (± 0.0088)** | 0.5638 (± 0.0133) | 0.5635 (± 0.0137) | 0.5636 (± 0.0136) | 0.5505 (± 0.0094) |
| robert | 0.5136 (± 0.0102) | 0.5134 (± 0.0119) | 0.5124 (± 0.0108) | **0.5159 (± 0.012)** | 0.51 (± 0.0098) |
| segment | 0.9442 (± 0.0141) | **0.9463 (± 0.0152)** | 0.9455 (± 0.0126) | 0.9442 (± 0.0139) | 0.9429 (± 0.0127) |
| shuttle | 0.8543 (± 0.009) | 0.8534 (± 0.0117) | 0.8534 (± 0.0117) | 0.8542 (± 0.009) | **0.9791 (± 0.0476)** |
| steel-plates-fault | **0.8491 (± 0.0211)** | 0.828 (± 0.0167) | 0.8234 (± 0.0198) | 0.8279 (± 0.0194) | 0.8167 (± 0.0197) |
| vehicle | 0.8519 (± 0.027) | 0.8589 (± 0.0297) | 0.8553 (± 0.0269) | 0.8552 (± 0.0264) | **0.8645 (± 0.0298)** |
| volkert | **0.7197 (± 0.0061)** | 0.6802 (± 0.0083) | 0.6795 (± 0.008) | 0.6805 (± 0.0086) | 0.6767 (± 0.0066) |
| wine-quality-white | **0.4279 (± 0.0438)** | 0.3938 (± 0.0417) | 0.3938 (± 0.0347) | 0.4023 (± 0.0333) | 0.3917 (± 0.0386) |
| yeast | **0.5213 (± 0.0644)** | 0.5113 (± 0.0556) | 0.5006 (± 0.0752) | 0.4955 (± 0.0732) | 0.516 (± 0.0569) |

Table 9: **ROC AUC - Binary**: The mean and standard deviation of the test score over all folds for each method. The best method per dataset is shown in bold.

| Dataset | CMA-ES | CMA-ES-ExplicitGES | GES | SingleBest | Stacking |
|---|---|---|---|---|---|
| APSFailure | 0.9923 (± 0.0021) | 0.9927 (± 0.0016) | **0.9927 (± 0.0016)** | 0.9925 (± 0.0017) | 0.992 (± 0.0015) |
| Amazon_employee_access | 0.901 (± 0.0125) | **0.9012 (± 0.0127)** | 0.9008 (± 0.0126) | 0.9003 (± 0.0119) | 0.8967 (± 0.0126) |
| Australian | 0.9399 (± 0.0189) | 0.9403 (± 0.0174) | 0.9403 (± 0.017) | 0.9402 (± 0.0187) | **0.9437 (± 0.0171)** |
| Bioresponse | 0.8843 (± 0.0178) | 0.8859 (± 0.0163) | **0.886 (± 0.016)** | 0.8807 (± 0.0166) | 0.8853 (± 0.016) |
| Click_prediction_small | 0.7098 (± 0.0116) | **0.7102 (± 0.0118)** | 0.7101 (± 0.0118) | 0.7086 (± 0.0122) | 0.71 (± 0.0119) |
| Higgs | **0.8256 (± 0.0008)** | 0.8244 (± 0.0008) | 0.8243 (± 0.0008) | 0.8244 (± 0.0008) | 0.8254 (± 0.0008) |
| Internet-Advertisements | 0.9859 (± 0.0106) | 0.9844 (± 0.0127) | 0.9851 (± 0.0129) | 0.9845 (± 0.0121) | **0.9866 (± 0.0106)** |
| KDDCup09-Upselling | 0.9085 (± 0.0067) | **0.9085 (± 0.0066)** | 0.9085 (± 0.0068) | 0.9082 (± 0.0067) | 0.8997 (± 0.0074) |
| KDDCup09_appetency | 0.8484 (± 0.0128) | 0.8487 (± 0.0131) | **0.8487 (± 0.0131)** | 0.8462 (± 0.0128) | 0.8373 (± 0.0129) |
| MiniBooNE | **0.9874 (± 0.0011)** | 0.9873 (± 0.001) | 0.9873 (± 0.001) | 0.9871 (± 0.001) | 0.9866 (± 0.0012) |
| PhishingWebsites | 0.9967 (± 0.0015) | 0.9969 (± 0.001) | **0.997 (± 0.001)** | 0.9955 (± 0.0015) | 0.9968 (± 0.001) |
| Satellite | 0.9682 (± 0.0883) | **0.9946 (± 0.0066)** | 0.9945 (± 0.0065) | 0.9944 (± 0.0066) | 0.9944 (± 0.006) |
| ada | 0.9199 (± 0.0174) | **0.9203 (± 0.0177)** | 0.9203 (± 0.0176) | 0.9198 (± 0.0179) | 0.9202 (± 0.0179) |
| adult | **0.9318 (± 0.004)** | 0.9316 (± 0.0041) | 0.9316 (± 0.0041) | 0.9312 (± 0.0044) | 0.9303 (± 0.0038) |
| airlines | 0.7064 (± 0.0024) | 0.7242 (± 0.0018) | **0.7245 (± 0.0019)** | 0.7233 (± 0.0019) | 0.7084 (± 0.0022) |
| albert | **0.7782 (± 0.0025)** | 0.778 (± 0.0025) | 0.7781 (± 0.0025) | 0.7776 (± 0.0026) | 0.7781 (± 0.0025) |
| arcene | **0.913 (± 0.1134)** | 0.8812 (± 0.1379) | 0.8812 (± 0.1379) | 0.8447 (± 0.1967) | 0.873 (± 0.1585) |
| bank-marketing | 0.9405 (± 0.0062) | 0.9406 (± 0.0061) | **0.9406 (± 0.0062)** | 0.9395 (± 0.0064) | 0.9399 (± 0.0062) |
| blood-transfusion-service-center | 0.7352 (± 0.059) | 0.7383 (± 0.0557) | 0.7394 (± 0.0552) | **0.7487 (± 0.0485)** | 0.7437 (± 0.0571) |
| christine | **0.8274 (± 0.0137)** | 0.8266 (± 0.0132) | 0.8266 (± 0.0134) | 0.8258 (± 0.0142) | 0.8274 (± 0.0135) |
| churn | **0.9348 (± 0.0164)** | 0.923 (± 0.0252) | 0.924 (± 0.0249) | 0.9221 (± 0.0247) | 0.929 (± 0.0195) |
| credit-g | 0.7926 (± 0.0329) | 0.797 (± 0.0367) | 0.7984 (± 0.0372) | 0.7894 (± 0.0323) | **0.7999 (± 0.0377)** |
| gina | **0.992 (± 0.0048)** | 0.9914 (± 0.0052) | 0.9914 (± 0.0052) | 0.991 (± 0.0057) | 0.9898 (± 0.0062) |
| guillermo | **0.9216 (± 0.0061)** | 0.9124 (± 0.0081) | 0.9119 (± 0.0077) | 0.9117 (± 0.0077) | 0.9214 (± 0.0058) |
| jasmine | 0.884 (± 0.0157) | 0.8856 (± 0.0165) | 0.8857 (± 0.0166) | 0.8836 (± 0.0178) | **0.8858 (± 0.0172)** |
| kc1 | 0.8338 (± 0.0409) | 0.8371 (± 0.0382) | 0.8378 (± 0.0367) | 0.8335 (± 0.0426) | **0.839 (± 0.0354)** |
| kick | 0.7913 (± 0.0062) | **0.7913 (± 0.0062)** | 0.7912 (± 0.0062) | 0.7898 (± 0.0057) | 0.7897 (± 0.0062) |
| kr-vs-kp | 0.9983 (± 0.0043) | **0.9998 (± 0.0002)** | 0.9998 (± 0.0002) | 0.9998 (± 0.0002) | 0.9994 (± 0.0012) |
| madeline | **0.9471 (± 0.0078)** | 0.9447 (± 0.0086) | 0.9447 (± 0.0087) | 0.9394 (± 0.0082) | 0.9458 (± 0.0094) |
| nomao | **0.9964 (± 0.0006)** | 0.9964 (± 0.0006) | 0.9964 (± 0.0006) | 0.9963 (± 0.0007) | 0.996 (± 0.0005) |
| numerai28.6 | 0.5301 (± 0.0045) | 0.5305 (± 0.0044) | **0.5305 (± 0.0045)** | 0.5297 (± 0.0044) | 0.5302 (± 0.0045) |
| ozone-level-8hr | 0.9267 (± 0.0287) | 0.9336 (± 0.0184) | **0.9338 (± 0.0177)** | 0.9329 (± 0.0193) | 0.9328 (± 0.0249) |
| pc4 | 0.9515 (± 0.0191) | **0.9526 (± 0.0191)** | 0.9524 (± 0.0195) | 0.9513 (± 0.0183) | 0.9519 (± 0.0191) |
| philippine | 0.877 (± 0.0129) | 0.8756 (± 0.013) | 0.8754 (± 0.0131) | **0.8772 (± 0.0117)** | 0.8754 (± 0.0132) |
| phoneme | **0.9717 (± 0.0087)** | 0.9684 (± 0.0094) | 0.9684 (± 0.0094) | 0.9678 (± 0.0091) | 0.9705 (± 0.0092) |
| porto-seguro | 0.5172 (± 0.0383) | **0.5172 (± 0.0382)** | 0.5172 (± 0.0382) | 0.5172 (± 0.0382) | 0.5172 (± 0.0381) |
| qsar-biodeg | 0.9398 (± 0.0313) | 0.9436 (± 0.029) | **0.9436 (± 0.0289)** | 0.9355 (± 0.0337) | 0.9418 (± 0.0307) |
| riccardo | **0.9999 (± 0.0001)** | 0.9998 (± 0.0001) | 0.9998 (± 0.0001) | 0.9997 (± 0.0002) | 0.9998 (± 0.0001) |
| sf-police-incidents | 0.6874 (± 0.0017) | 0.6886 (± 0.0019) | **0.6886 (± 0.0019)** | 0.6873 (± 0.0017) | 0.684 (± 0.0019) |
| sylvine | 0.9863 (± 0.0071) | **0.9889 (± 0.0033)** | 0.9889 (± 0.0034) | 0.988 (± 0.0039) | 0.9884 (± 0.0036) |
| wilt | 0.994 (± 0.0071) | 0.9946 (± 0.0091) | **0.9948 (± 0.0087)** | 0.9946 (± 0.0092) | 0.9943 (± 0.0084) |

Table 10: **ROC AUC - Multi-class**: The mean and standard deviation of the test score over all folds for each method. The best method per dataset is shown in bold.

| Dataset | CMA-ES | CMA-ES-ExplicitGES | GES | SingleBest | Stacking |
|---|---|---|---|---|---|
| Diabetes130US | 0.7127 (± 0.0054) | **0.713 (± 0.0055)** | 0.713 (± 0.0055) | 0.7129 (± 0.0055) | 0.7115 (± 0.0053) |
| Fashion-MNIST | 0.9942 (± 0.0004) | **0.9943 (± 0.0004)** | 0.9943 (± 0.0004) | 0.9941 (± 0.0004) | 0.9937 (± 0.0005) |
| GesturePhaseSegmentationProcessed | **0.94 (± 0.0066)** | 0.9382 (± 0.0064) | 0.938 (± 0.0064) | 0.9374 (± 0.0063) | 0.937 (± 0.0074) |
| KDDCup99 | **0.9999 (± 0.0001)** | 0.8898 (± 0.0233) | 0.8935 (± 0.0148) | 0.892 (± 0.0174) | 0.9997 (± 0.0005) |
| amazon-commerce-reviews | **0.995 (± 0.0019)** | 0.994 (± 0.0024) | 0.9941 (± 0.0022) | 0.9924 (± 0.0037) | 0.9919 (± 0.0021) |
| car | 0.9993 (± 0.0022) | 1.0 (± 0.0001) | 1.0 (± 0.0001) | 1.0 (± 0.0001) | **1.0 (± 0.0)** |
| cmc | 0.7353 (± 0.0338) | 0.739 (± 0.0345) | **0.7391 (± 0.0344)** | 0.7296 (± 0.0332) | 0.7358 (± 0.0344) |
| cnae-9 | 0.9974 (± 0.0035) | **0.9985 (± 0.0018)** | 0.9985 (± 0.0017) | 0.9983 (± 0.0018) | 0.9984 (± 0.0016) |
| connect-4 | 0.9494 (± 0.0035) | **0.9496 (± 0.0035)** | 0.9496 (± 0.0035) | 0.9496 (± 0.0035) | 0.9472 (± 0.0034) |
| covertype | 0.9994 (± 0.0) | **0.9995 (± 0.0)** | 0.9995 (± 0.0) | 0.9994 (± 0.0) | 0.9993 (± 0.0001) |
| dilbert | 0.9999 (± 0.0001) | 0.9999 (± 0.0) | 0.9999 (± 0.0) | 0.9999 (± 0.0002) | **1.0 (± 0.0)** |
| dionis | 0.9937 (± 0.0027) | 0.9941 (± 0.0025) | 0.9941 (± 0.0025) | 0.9891 (± 0.0051) | **0.997 (± 0.0004)** |
| dna | 0.9943 (± 0.0033) | 0.9952 (± 0.0025) | **0.9953 (± 0.0026)** | 0.9947 (± 0.0026) | 0.9947 (± 0.003) |
| eucalyptus | 0.9293 (± 0.0173) | 0.9321 (± 0.0154) | 0.9314 (± 0.017) | 0.9296 (± 0.0142) | **0.9347 (± 0.0156)** |
| fabert | 0.9457 (± 0.0038) | **0.946 (± 0.0038)** | 0.9458 (± 0.0038) | 0.9445 (± 0.004) | 0.9434 (± 0.0044) |
| first-order-theorem-proving | 0.8468 (± 0.0114) | 0.8523 (± 0.0094) | **0.8524 (± 0.0095)** | 0.8468 (± 0.0109) | 0.8408 (± 0.0115) |
| helena | 0.8992 (± 0.0027) | 0.8999 (± 0.0027) | **0.9 (± 0.0027)** | 0.8986 (± 0.0028) | 0.8795 (± 0.0014) |
| jannis | 0.8872 (± 0.0032) | **0.8887 (± 0.0031)** | 0.8887 (± 0.0031) | 0.8872 (± 0.0032) | 0.8846 (± 0.0027) |
| jungle_chess_2pcs_raw_endgame_complete | **0.9992 (± 0.0003)** | 0.999 (± 0.0002) | 0.999 (± 0.0002) | 0.999 (± 0.0002) | 0.9991 (± 0.0003) |
| mfeat-factors | 0.9989 (± 0.0017) | **0.9996 (± 0.0004)** | 0.9995 (± 0.0005) | 0.9994 (± 0.0007) | 0.9992 (± 0.0007) |
| micro-mass | 0.9913 (± 0.0167) | 0.9956 (± 0.008) | 0.9956 (± 0.0082) | 0.9957 (± 0.0081) | **0.9976 (± 0.002)** |
| okcupid-stem | 0.8321 (± 0.0059) | 0.8329 (± 0.0055) | **0.8329 (± 0.0055)** | 0.832 (± 0.0057) | 0.8277 (± 0.005) |
| robert | **0.8866 (± 0.0036)** | 0.8853 (± 0.0039) | 0.8855 (± 0.004) | 0.8843 (± 0.0041) | 0.8817 (± 0.005) |
| segment | 0.9964 (± 0.0009) | **0.9965 (± 0.0013)** | 0.9965 (± 0.0012) | 0.9961 (± 0.0014) | 0.9962 (± 0.0012) |
| shuttle | 1.0 (± 0.0) | 0.9286 (± 0.0) | 0.9286 (± 0.0) | 0.9286 (± 0.0) | **1.0 (± 0.0)** |
| steel-plates-fault | 0.9663 (± 0.0086) | **0.9696 (± 0.0048)** | 0.9694 (± 0.0049) | 0.9678 (± 0.0062) | 0.9653 (± 0.0088) |
| vehicle | 0.9665 (± 0.0102) | **0.9691 (± 0.0091)** | 0.9686 (± 0.0092) | 0.9689 (± 0.0086) | 0.9683 (± 0.0091) |
| volkert | 0.9569 (± 0.0011) | **0.9581 (± 0.001)** | 0.9581 (± 0.001) | 0.9578 (± 0.0011) | 0.9521 (± 0.0017) |
| wine-quality-white | 0.8594 (± 0.0306) | 0.8409 (± 0.0319) | 0.8425 (± 0.0311) | 0.8446 (± 0.0311) | **0.8673 (± 0.0339)** |
| yeast | **0.8834 (± 0.035)** | 0.8632 (± 0.0334) | 0.8638 (± 0.0327) | 0.8571 (± 0.0384) | 0.878 (± 0.0325) |

