# OpenReview forum: "CMA-ES for Post Hoc Ensembling in AutoML: A Great Success and Salvageable Failure"
_automl.cc/AutoML/2023/Conference — AutoML 2023 MainTrack_

### Review · Reproducibility_Reviewer_tftS · 2023-04-07

**Completeness Of Code And Dataset Supplement Rating:** 2
**Usability And Ease Of Reproducibility Rating:** 2

**Actions Required To Increase The Reproducibility And Overall Recommendation:**

Please do the following:

* Either provide a `requirements.txt` + `setup.py` file, or please give exact instructions on how to load the Docker environment file.
* Provide the implementation of the CMA-ES algorithm used throughout the paper.

**Completeness Of Code And Dataset Supplement:**

(See below on my attempts to actually run the code, which also cuased problems).

In terms of approximate code completeness, it seems that the code provided seems to perform some form of data retrieval from e.g. OpenML / Autogluon.

However, there is no code for the exact implementation of CMA-ES or GES that the paper mentions throughout (let alone hyperparameters, etc.). Thus I would say the code is very incomplete at the moment for reproducing any of the paper's results.

**Overall Reproducibility Review:**

The code appears to be well organized and does provide specific commands, but has incredibly ambiguous installing documentation (no precise instructions on using the Docker file). The code also has very strangely worded variables/objects that make it difficult to understand their purpose. Several key algorithm components (e.g. CMA-ES implementation) are completely lacking, and even the paper itself does not describe their precise hyperparameters (i.e. What are "default" hyperparameters?)

**Review Confidence:**

4: You are confident in your assessment, but not absolutely certain. It is unlikely, but not impossible, that you did not understand some parts of the submission or that you are unfamiliar with some pieces of the code or data.

**Review Rating:**

3: Reject, you were not able to reproduce some critical aspects of the paper, and question whether it would be possible even with additional effort.

**Review Summary:**

I vote for reject currently - the reproducibility of the paper overall is poor.

**Summary Of Necessary Code And Dataset Supplement:**

This paper in summary tries to optimize a "user-defined loss" objective, where the search space is the set of vectors $w = (w_{1},...,w_{m})$ which sum to 1. The weight vector $w$ represents the ensembling weight among the outputs of $m$ different models (whose data is collected from the AutoML benchmark).

**Usability And Ease Of Reproducibility:**

I was unable to run the code, primarily due to the paper's lack of concise documentation on running Docker (only a link was provided to the general Docker website). This type reproducibility repository is very abnormal - normally repositories should simply have a `setup.py` in addition to `requirements.txt`, to allow a simple `pip install`. I think it's unreasonable to expect a regular researcher to immediately know how to use this Docker software (without any specific instructions).

Docker requires installing an entire application (> 1GB), and even then, there was no direct instruction on how to load the `DockerfileCollector` provided. I tried searching up online, e.g. https://www.linode.com/docs/guides/how-to-use-dockerfiles/ but no commands worked so far.

Furthermore, it's unclear what the code is actually doing many times, especially due to the naming of files and objects (e.g. what is an "assembler"? What is a "meta task"? - these were not defined in the paper either).

---

> ### Author Response · Authors · 2023-04-24
> **Response to Reproducibility Review of tftS**
>
> Dear Reviewer tftS,
>
> We would like to respond to several points you have raised and help you to reproduce the code. We have already updated the code base on anonymous GitHub with the minor changes mentioned below.
>
> > “However, there is no code for the exact implementation of CMA-ES or GES that the paper mentions throughout (let alone hyperparameters, etc.). Thus I would say the code is very incomplete at the moment for reproducing any of the paper's results.”
>
> We are baffled how you could come to this conclusion. The code contains the implementation of CMA-ES and GES (even before our update today, as the Code And Dataset Supplement .ZIP file on OpenReview can prove). There is an entire directory for ensembling methods, immediately mentioned in the README. In detail, you can find the methods you are looking for under `assembled_ensembles/methods/numerical_solvers/cmaes.py` and `assembled_ensembles/methods/ensemble_selection/greedy_ensemble_selection.py`.
>
> > “Most of the filled responses in the reproducibility checklist were simply "See the code."”
>
> This is the intended way of filling the checklist as described in the template of the AutoML conf (see, https://github.com/automl-conf/LatexTemplate/blob/main/instructions.tex#L327-L332)
>
> > “Another response 3(n) on hyperparameters was responded with "We did not tune hyperparameters. We used a default application of CMA-ES and introduced no meaningful new hyperparameters with our approaches that would require tuning." which is incredibly vague, especially since the implementation of CMA-ES was not provided and "default" hyperparameters are relative to the implementation.”
>
> As we provided code for the method, this point should resolve itself once you find the code.
>
> > I was unable to run the code, primarily due to the paper's lack of concise documentation on running Docker (only a link was provided to the general Docker website). This type reproducibility repository is very abnormal - normally repositories should simply have a `setup.py` in addition to `requirements.txt`, to allow a simple `pip install`. I think it's unreasonable to expect a regular researcher to immediately know how to use this Docker software (without any specific instructions).
>
> Docker is not required for our repository, as stated in our repository. Nevertheless, it is a very common tool for Python and has many online tutorials on how to use it. We did not deem it necessary to explain in the same way we did not deem it necessary to explain how to install Python.
>
> To ease the use without docker, we now provide a `requirements.txt` in the repository besides the docker and singularity files for the second step – the last step already included a `requirements.txt` and can be tested independently from the two other steps as described in the README. The new `requirements.txt` file contains the contents of the docker files. Furthermore, we provided more details on the installation in the readme of the steps.
>
> For the first step, we detailed specific CLI-based instructions to obtain the same environment -- one-to-one following the docker file. We can not provide a `requirements.txt` for this step because the `requirements.txt`-based install does not support specific `pip` options that our installation requires.
>
> We still highly recommend using docker instead of Python, which guarantees that you are using exactly the same system we used. If this is impossible, the new `requirements.txt`- and CLI-based installation also works. We are also not sure what you referring to with `DockerfileCollector`. Could you clarify this or verify if the `requirements.txt`- and CLI-based installation help you to reproduce the code?
>
> > “Furthermore, it's unclear what the code is actually doing many times, especially due to the naming of files and objects (e.g. what is an "assembler"? What is a "meta task"? - these were not defined in the paper either).”
>
> You are referring to class names. These are simply names we selected for objects during our implementation and have no direct connection (in terms of meaning) to the paper. Each class has doc strings and should help you to answer some of your questions. Moreover, there is the documentation for the frameworks we built upon (e.g. for assembler, it's `AutoGluon`; for metatasks, it's `assembled`), which should also help you to answer some of your questions that are not directly related to our code.
>
> &nbsp;
>
> In case you have any further questions regarding the code, please reach out and reply to this comment so that we can help you.
> We deem our code highly reproducible and have not encountered any problems related to using docker or singularity.

---

> > ### Comment · Area_Chair_Vqah · 2023-05-08
> > **Reproducibility reviewer?**
> >
> > Dear Reviewer tftS,
> >
> > Could you comment on the answer given by the authors to your review? At the moment, there is a strong disagreement between the two of you, so it would be good if we could converge (or make precise what exactly the disagreement is).
> >
> > All the best
> > Your area chair

---

> > ### Comment · Reproducibility_Reviewer_tftS · 2023-05-10
> > **Thanks for clarification**
> >
> > Thank you for the clarification and apologies for the delay. As for updates:
> >
> > * I now see the `cmaes.py` file as pointed out from the authors' response, although this required looking at the upper-level `README.md` which only points to `/assembled_ensembles/README.md`.
> > * I also have found the new `requirements.txt` (IMO this should be at the beginning of the repository, not the `/evaluation/` folder).
> > * As for the class names, if the documentation (i.e. subfolder READMEs) points out their function, then this appears satisfactory.
> >
> > I thus wish to update my score to "weak accept" reflect these changes. The reason I do not push for stronger is because from a software engineering perspective, the code still lacks very clean organization. I overall would recommend in the future to have a more concise way of conveying the code however, to avoid such kinds of issues again (e.g. a user cannot easily find certain files because they are strangely located).

---

### Official Review · Reviewer_Bj9B · 2023-04-11

**Potential Impact On The Field Of Automl Rating:** 4
**Technical Quality And Correctness Rating:** 2
**Clarity Rating:** 3

**Summary Of Contributions:**

This paper studies the problem of post-hoc ensembling in AutoML systems, and proposes schemes to improve upon the widely used Greedy Ensemble Selection or GES scheme. To this end, this paper considers the use of CMA-ES for ensemble selection in place of GES by posing the ensemble selection as a gradient-free numerical unconstrained optimization of the ensemble weights using validation data.

The paper empirically evaluates this use of CMA-ES for ensemble selection on AutoML classification problems, with the results highlighting that CMA-ES is able to improve upon GES when considering the balanced accuracy metric, but falls behind GES when considering the ROC-AUC metric. Based on these results, the paper then motivates the need for constraints on the ensemble weights during the numerical optimization with CMA-ES, and presents three ways of enforcing such constraints. The empirical evaluation of these constraint enforcement schemes demonstrates that CMA-ES with such constraints improves upon vanilla CMA-ES, and is able to match GES for the ROC-AUC metric. However, these constraints lead to performance degradation with the balanced accuracy metric.


**Actions Required To Increase Overall Recommendation:**

I am happy to increase my score if the authors (or other reviewers) can point out how my concerns regarding the fundamental issues with the proposed idea is not justified or if it is a result of my misunderstanding of the idea.

Alternately, if we can understand some fundamental reasons why certain ensembling strategies or weighing schemes help with certain metrics but can hurt other metrics, it would be a valueable contribution as it will improve the understanding of the community. But this might be out of scope within a review cycle.


**Clarity:**

The paper is very well written and easy to follow in most places. The authors provide ample motivation for the various choices and evaluations made in the paper. For example, the explicit explanation of the "pseudo-discrete" and "sparse" properties of GES are very intuitive.

There are couple of aspects that I believe can be better clarified:

- First, it would be useful to present an extended version of the Tables 1 and 2 where the rank changes for all the methods are presented. It is understood that if CMA-ES-*'s rank is improving, the ranks of others must be degrading. But it would be useful to see this detail explicitly. A somewhat related issue is that it is not clear why the "absolute rank" in Table 2 can be 1.5.

- Second, it might be useful to present the precise steps in the weight normalization schemes more clearly, for example, in an algorithm environment. If space is an issue, having it in the appendix would have been useful.

- Finally, it would be good to better motivate the use of CMA-ES compared to other schemes such as Bayesian optimization (there are various schemes for high-dimensional Bayesian optimization if $m$ is large). The authors mention that CMA-ES is a  "state-of-the-art gradient-free numerical optimization" scheme. However, it would be good to have some support for this claim, along with some intuition as to what it is about CMA-ES that makes it  suitable for ensemble selection. For example, GES has an intuitive greedy search interpretation.

- [minor] The intro section has no header or number


**Overall Review:**

### Positive Aspects

Post-hoc ensembling is a widely used technique in many AutoML toolkits. Thus, any significant improvement to this ensembling scheme can have significant practical impact. This paper demonstrates that CMA-ES is able to significantly improve over GES on the AMLB when considering the balanced-accuracy benchmark.


The proposed method is clearly motivated and thoroughly evaluated with the AutoML benchmark (AMLB) with 71 classification datasets with two classification metrics. The results are presented with statistical significance tests, making it suitable for drawing meaningful conclusions from the empirical evaluations. Such an elaborate evaluation is critical when addressing fundamental aspects of AutoML such as post-hoc ensembling.

The paper does a great job at highlighting the desirable properties of GES and how to incorporate such properties as constraints with CMA-ES. This was very clearly articulated.



### Negative Aspects

I have detailed my main technical concerns with this paper in the **Technical Quality and Correctness** section of the review. Among those, the main concern is the fact that the modifications made to CMA-ES to improve performance with ROC-AUC metric leads to degradation in the performance with the balanced-accuracy metric. This gives the impression that we are not clearly understanding why we are seeing improvements or degradation over GES, and that there is some implicit aspect we are missing. Some of the other technical issues such as (i) small number of base models, or (ii) the use of AutoGluon base models, or (iii) the thresholding effect with balanced accuracy, or (iv) the fact that SingleBest performs competitively to GES with balanced accuracy might all be related to the main concern that CMA-ES enhancements for ROC-AUC hurt the balanced accuracy performance.
It is not clear if there is any fundamental improvement that CMA-ES provides over GES in a metric-agnostic way. One fundamental aspect could have been that CMA-ES allows for negative weights, while GES restricts to non-negative weights. But the CMA-ES normalizations remove this distinguishing feature, implicitly implying that the negative weights hurt.
Another related issue is that, if the CMA-ES needs to be configured differently for different metrics to get improvements over GES (or even match it), then its usability is probably going to be somewhat limited relative to GES. Would this mean that if we consider other metrics (such as F1-score, or area under the precision-recall curve), we would again have to reconfigure CMA-ES ensembling?


**Potential Impact On The Field Of Automl:**

Post-hoc ensembling is a widely used technique in many AutoML toolkits. Thus, any significant improvement to this ensembling scheme can have significant practical impact.


**Review Confidence:**

4: You are confident in your assessment, but not absolutely certain. It is unlikely, but not impossible, that you did not understand some parts of the submission or that you are unfamiliar with some pieces of related work.

**Review Rating:**

7: Weak Accept: Technically sound paper with moderate-to-high impact and strong evaluation, with perhaps some minor flaws.

**Review Summary:**

The paper is studying a very important problem and provides an elaborate evaluation worthy of current AutoML research. However, I feel that there are certain fundamental limitations (as described in detail) which leads me to recommend a rejection.


**Technical Quality And Correctness:**

The proposed method is clearly motivated and thoroughly evaluated with the AutoML benchmark (AMLB) with 71 classification datasets with two classification metrics. The results are presented with statistical significance tests, making it suitable for drawing meaningful conclusions from the empirical evaluations.


However, I have a few concerns with some of the technical aspects:

**Small number of base models.**
First, the mean number of base models for all the datasets and experiments appear to be very small -- in the twenties or less. My understanding is that usually we would pick the top-$m$ (with $m = 50$ default in auto-sklearn) of all the different models created. I wonder if such a small number of base models in the ensemble have an effect of the conclusions made.

**Thresholding effect.**
Second, if we are seeing a consistent improvement for balanced accuracy but no improvement for ROC-AUC, then it feels like that there might be some implicit effects of threshold calibration. For ROC-AUC, there is no threshold (as mentioned by the authors), but we also do not see improvements with CMA-ES. With (balanced) accuracy, it almost seems like the CMA-ES ensembling, which potentially can contain more ensemble members than GES (as mentioned in lines 45-46), is having some implicit calibration effects that allow for improved balanced accuracy at the default classification threshold of 0.5. In this case, it is not entirely clear to me why CMA-ES is better. A potential way of teasing out the effect of the different ensembling schemes can be the following -- for each scheme and dataset, we can again use the validation data (or out-of-fold data) to calibrate the probability threshold by picking the threshold that maximizes the balanced accuracy. Then we can compare the calibrated GES performance to the calibrated CMA-ES performance to explicitly ablate the effect of the ensembling scheme.

**SingleBest vs GES.**
Another technical concern is that the results in Figure 1 (and Section 2) seems to indicate that there is not a lot of difference between SingleBest and GES (for the balanced accuracy metric), which seems to contradict the whole premise of ensembling the top-$m$ base models instead of just using the SingleBest. This seems a bit concerning given that existing papers have shown that GES can significantly improve over the SingleBest. What am I missing here? Might this be related to the fact that the number of base models for each dataset in these experiments is quite small?

**CMA-ES with normalization hurts balanced accuracy performance.**
Furthermore, while the normalization of the weights is very well motivated, the results leave a lot of questions that are not discussed. For example, if "pseudo-discrete" and "sparse" properties are important, we should try to ablate the effect of each of these properties. Do the 3 variations exhibit such an ablation? All normalization methods seem to have similar performance in Figure 4. So it is not clear how much improvement we are seeing over CMA-ES. Furthermore, the performance of CMA-ES with normalization seems to be significantly worse than that of CMA-ES with balanced accuracy. Does this mean that we would have to utilize different versions of CMA-ES ensembling for different metrics to get the best ensembling? This seems counter-intuitive. On a related note, there is no discussion why the GES properties help CMA-ES with ROC-AUC but hurt CMA-ES with balanced accuracy. What might be happening in the ensembling that leads to such different behaviours with different metrics?


**Use of AutoGluon base models.**
Finally, one thing that is unclear is the choice of AutoGluon as the base model generator. AutoGluon is a very powerful AutoML tool with very strong performance compared to other open-source tools. However, to the best of my knowledge (which might be limited given AutoGluon goes through updates), the base model choices are somewhat pre-specified, and the main gain in AutoGluon comes from various forms of ensembling -- the $k$-fold ensembling, the across-model ensembling, the stacking -- and the anytime strong performance is achieved via careful ordering of how the different base models in the ensemble are generated given the time budget. Unless HPO is enabled (which it is not by default), the set of base models are somewhat data independent, implying that different datasets would get the same set of base models if the model training took the same amount of time for the different datasets. This is not necessarily invalidating the results presented in this paper, but it is a bit of a different setup than the usual post-hoc ensembling where the set of base models themselves are often data dependent.

---

> ### Author Response · Authors · 2023-04-24
> **Response to Review of Bj9B (Part 1/2)**
>
> Dear Reviewer Bj9B,
>
> Thank you for your feedback. The following replies to your concerns. Thereby, directly answering some of your concerns raised under negative aspects. As we want to start a potential discussion early, given the short timeframe of the rebuttal, we are sharing our comments now and are stating below where we plan to adjust the manuscript. The updated manuscript will then follow in the next few days.
>
>
> ## Technical Quality And Correctness
> ### Small number of base models
> First, to clarify, the number of base models is entirely controlled by AutoGluon.
> AutoGluon only produces a small number of base models in most cases, because it expensively trains and evaluates a configuration with n-repeated k-fold cross-validation and only has a small number of possible configurations in total. In contrast, an AutoML system, like Auto-Sklearn, uses a less expensive validation procedure (e.g., hold-out) to train and evaluates much more base models during Bayesian optimization (>>50). Moreover, such systems decided to prune to a smaller number of base models (e.g. 50); details on the reasons for this are missing in papers of AutoML systems.
>
> We agree that this prompts an interesting research question. It would require detailed analysis, likely with additional AutoML systems, to determine the effect of fewer base models on post hoc ensembling. This is, however, out of the scope of this rebuttal. We will note in the limitations that we only evaluated our approach for one AutoML system with its specific properties and that we make no guarantees for AutoML systems with entirely different approaches to CASH.
>
> Nevertheless, we think the smaller number of base models has no significant effect on our conclusion because the results of GES for ROC AUC indicate that the statement made for Auto-Sklearn, which has a large number of base models, holds – unaffected by the smaller number of base models and the entirely different approach of AutoGluon to CASH in AutoML. Moreover, AutoGluon uses GES, following Auto-Sklearn – despite the fundamental differences.
>
> ### Thresholding effect
> We would like to note that any effect that functions like calibration/changes to the threshold only holds for binary classification, as no compared ensemble method performs a schema for multi-class that is suitable for calibration (e.g., One-Vs-Rest classification). Nevertheless, we see the same trends for CMA-ES across binary and multi-class classification.
>
> Moreover, we hypothesize that post hoc ensembling methods benefit from performing something like calibration in binary classification. They reweight the impact of the probabilities such that the base models’ threshold of 0.5 might move given the ensemble’s static threshold of 0.5. But this is not exclusive to CMA-ES, GES achieves the same reweighting of base models.
>
> Hence, we do not think that the thresholding effect impacts our work or conclusions.
>
> ### SingleBest vs GES (for balanced accuracy)
>
> You stated:
> > “This seems a bit concerning given that existing papers have shown that GES can significantly improve over the SingleBest.”
>
> Which papers are you referring to? We are only aware of papers stating this, but no paper has ever shown this. Moreover, AutoML systems are usually also not evaluated on balanced accuracy (see the AutoML benchmark), which is likely why previous papers did not consider this in their statements.
>
> Figure 1 shows the mean rank, which is drastically dominated by CMA-ES such that GES and the SingleBest have the same mean rank in the end. The relative performance comparison (Figure 3 a/b), shows the (marginal) improvement of GES over the SingleBest more clearly (especially for multi-class).
>
> We see no connection to the number of base models for this point. If this would be a reason, CMA-ES would be equally affected by it. Nevertheless, we would be thankful if you could elaborate on the last question.
>
> ## CMA-ES with normalization hurts balanced accuracy performance
>
> > “For example, if "pseudo-discrete" and "sparse" properties are important, we should try to ablate the effect of each of these properties. Do the 3 variations exhibit such an ablation?”
>
> Yes, the variations exhibit such ablation, but not explicitly. CMA-ES and CMA-ES-Softmax are versions without either constraint; CMA-ES-ImplicitGES satisfies “sparse”; and CMA-ES-ExplicitGES satisfies both. Moreover, as Appendix E shows, only the version that correctly satisfies both properties (CMA-ES-ExplicitGES), is significantly different from CMA-ES for multi-class and always has the highest mean rank. We will provide more details on this in Appendix E.

---

> > ### Comment · Reviewer_Bj9B · 2023-04-26
> > **Paper on balanced accuracy**
> >
> > I am still digesting the response, but in the meantime here is the paper regarding the following comment:
> >
> > > “This seems a bit concerning given that existing papers have shown that GES can significantly improve over the SingleBest.”
> > >
> > > Which papers are you referring to? We are only aware of papers stating this, but no paper has ever shown this. Moreover, AutoML systems are usually also not evaluated on balanced accuracy (see the AutoML benchmark), which is likely why previous papers did not consider this in their statements.
> >
> > The original auto-sklearn paper [R1] uses balanced error rate and shows that ensembling improves upon SingleBest -- for example see Figure 3 in the paper. Note that they compare 4 algorithms in this figure so the difference between vanilla auto-sklearn and ensembled auto-sklearn looks small but is significant on a 1-1 comparison.
> >
> > To the best of my understanding, balanced error rate averages the per class error rate, and hence, is just 1 - balanced accuracy. So this is equivalently a result on balanced accuracy. Of course this evaluation pre-dates the AMLB.
> >
> > [R1] Feurer et al. 2015. [Efficient and Robust Automated Machine Learning](https://papers.nips.cc/paper_files/paper/2015/hash/11d0e6287202fced83f79975ec59a3a6-Abstract.html). NeurIPS.

---

> > > ### Author Response · Authors · 2023-04-27
> > > **Response: Paper on balanced accuracy**
> > >
> > > We agree that the original auto-sklearn paper shows that GES improves over the SingleBest. Moreover, we agree that the balanced error rate is just 1 - balanced accuracy. But we disagree that the original auto-sklearn paper shows a **statistically** significant improvement; which we thought was what you are referring to.
> > >
> > > We believe it is hard to tell from an average relative rank to conclude significance on a 1-1 comparison. Any of the other 2 methods could have been better than auto-sklearn + ensemble and thus worsen its average rank, but similarly, the other 2 methods could have also only been better than vanilla auto-sklearn and worsened its average rank.
> > >
> > > Therefore, we took the time to compare vanilla auto-sklearn and auto-sklearn + ensemble (the original data for Figure 3 can be found [here](https://figshare.com/articles/dataset/Efficient_and_Robust_Automated_Machine_Learning_-_Section_6/3824103)).
> > > We computed the average rank over all 140 datasets when only comparing vanilla auto-sklearn and auto-sklearn + ensemble: vanilla auto-sklearn has a mean rank of $1.59$ and auto-sklearn + ensemble of $1.41$. Comparing the performances with a Wilcoxon Test (scipy's default implementation) returns a p-value of $0.095$; assuming a standard $\alpha = 0.05$, **not significant**.
> > >
> > > To additionally put our results into context, we computed the average rank of a 1-1 comparison of GES and SingleBest for our data of AutoGluon: GES ranks marginally better than SingleBest for balanced accuracy binary ($1.49$ vs. $1.51$; 41 datasets) and multi-class ($1.47$ vs. $1.53$; 30 datasets).
> > >
> > > Considering that many of the datasets used in the original auto-sklearn paper are not considered good benchmark tasks anymore (e.g., following the requirements of the latest AutoML benchmark paper [R2]), we believe the results to be similar enough (in terms of rank and significance) to contradict your original statement that our results are concerning.
> > >
> > > Finally, we note that for subsequent benchmarks and comparisons of AutoML systems after the original auto-sklearn paper, balanced error rate/accuracy has not been evaluated again [R2, R3, R4, R5] -- which we believe to be a mistake given the differences in performance between ROC AUC and balanced accuracy found in our paper (for AutoGluon). In contrast, the latest auto-sklearn paper [R6] reports (normalized) balanced error rates for ablation and portfolio/meta-learning studies but not in comparison to other AutoML systems.
> > >
> > > &nbsp;
> > >
> > > References
> > > * [R1] Feurer et al. 2015. Efficient and Robust Automated Machine Learning. NeurIPS.
> > > * [R2] AutoML Benchmark Papers https://openml.github.io/automlbenchmark/papers.html
> > > * [R3] AutoMLBench: A Comprehensive Experimental Evaluation of Automated Machine Learning Frameworks, https://arxiv.org/abs/2204.08358
> > > * [R4] Can AutoML outperform humans? An evaluation on popular OpenML datasets using AutoML Benchmark, https://dl.acm.org/doi/abs/10.1145/3448326.3448353
> > > * [R5] AutoGluon-Tabular: Robust and Accurate AutoML for Structured Data, https://arxiv.org/abs/2003.06505
> > > * [R6] Auto-Sklearn 2.0: Hands-free AutoML via Meta-Learning, https://arxiv.org/abs/2007.04074

---

> > > > ### Comment · Reviewer_Bj9B · 2023-05-01
> > > > **Thank you for the response**
> > > >
> > > > Thank you for the thorough point by point response, the new empirical studies. I am happy to raise my score. The paper is covering an important topic and it makes sense to explore how we can improve over the GES, which is what this paper does. The updates in the paper make the paper much more clear, and the additional results in the supplement makes it clear how the comparison is done (beyond the summaries in the main paper).
> > > >
> > > > I also really appreciate the investigation into the autosklearn's comparison between SingleBest vs Ensemble. I wonder if the difference between warm-started autosklearn and warm-started+ensembled autosklearn is also similarly statistically insignificant.
> > > >
> > > >
> > > > However I continue to have some reservations regarding the exploration considered in this paper:
> > > > - First, the small number of base models considered for the evaluation seems a bit incomplete. I understand that this number is completely controlled by autogluon but our goal here is not the evaluate autogluon but rather a subset selection algorithm. A more complete evaluation should be considering performance with varying numbers of base models. However, such an evaluation might not be feasible in the review period (especially since I was not able to respond earlier).
> > > > - The other reservation I have is the metric-dependent "overfitting" of CMAES. The results highlight that, with CMAES, the validation objective is much better than the test objective with auroc and thus we say CMAES overfits. But I don't think we have an understanding of why this phenomenon metric dependent. Shouldn't the proposed regularization help with balanced accuracy too? Why is there no overfitting with unconstrained CMAES on balanced accuracy? I don't feel like we have a good answer here. The authors claim that this phenomenon might be dependent on whether the metric is threshold based, and the response says that CMAES might do well other threshold based metrics. But that hypothesis needs to be supported. And similarly, it would good to show that unconstrained  CMAES also "overfits" with area under the precision-recall curve metric (much like AuROC). Without these supporting evidence (or some mathematical analyses), we don't really understand what is happening here. I don't think just claiming no free lunch addresses this issue.
> > > >
> > > >
> > > > As a final comment, it would be great if the authors can point me to the autogluon paper that talks about it using GES. To the best of my understanding (of some of the papers and the various presentations), autogluon just ensembles everything they train during the given time period, with a careful (preset) selection of what gets trained when. Of course, there is a paper about how such a large model is distilled into a smaller model. But I was not aware of there being any base models selection in the ensemble.

---

> > > > > ### Author Response · Authors · 2023-05-01
> > > > > **Responses: Additional Concerns Reviewer Bj9B**
> > > > >
> > > > > Thank you for your response and reconsideration. The following provides our answers to your additional concerns.
> > > > >
> > > > > > "I wonder if the difference between warm-started autosklearn and warm-started+ensembled autosklearn is also similarly statistically insignificant."
> > > > >
> > > > > Here are the results, as this only requires changing paths in our code: warm-started auto-sklearn $1.55$ vs.  warm-started+ensembled auto-sklearn $1.45$ (Wilcoxon $0.087$). Note, the (comparably) poor performances of warm-started+ensembled auto-sklearn and auto-sklearn + ensemble likely follow from overfitting due to hold-out validation. On the validation scores, they are significantly different, and the SingleBest is heavily outranked.
> > > > >
> > > > >
> > > > > > "First, the small number of base models considered for the evaluation seems a bit incomplete. I understand that this number is completely controlled by autogluon but our goal here is not the evaluate autogluon but rather a subset selection algorithm. A more complete evaluation should be considering performance with varying numbers of base models."
> > > > >
> > > > > We agree that such an evaluation would be very insightful and clarify the significance of the impact of a small to large number of base models on overfitting. However, we believe that modifying the base models produced by AutoGluon (or any other AutoML system) would skew our result. Our goal was to evaluate a post hoc ensembling method for a state-of-the-art AutoML system with a good validation procedure. Therefore, following best practices (cf. AutoML benchmark, Auto-Sklearn papers, AutoGluon paper), we had to evaluate AutoML system with a time limit. This time limit is directly responsible for the number of base models, as AutoGluon (or any other AutoML system with such an expensive validation procedure) simply often won't have the time to validate more models or enough models such that we can freely choose the number of base models for our experiments. Therefore, we believe that post hoc ensembling method (i.e., GES with its weighting and selection or default CMA-ES with only weighting) must cope with the results of a time-limited AutoML system to draw realistic conclusions about their performance. Nevertheless, in future work, we would like to explore this in a "lab-like" setting where we can control the number of base models and ignore time constraints.
> > > > >
> > > > >
> > > > >
> > > > > > "But I don't think we have an understanding of why this phenomenon metric dependent. [..]. I don't think just claiming no free lunch addresses this issue."
> > > > >
> > > > > We agree that our work does not provide a complete answer to this question. Following the goal of our paper, our work started by trying to verify the statements of auto-sklearn for another AutoML system with a better validation procedure and ended up opening many new future research opportunities. Consequently, our work provides, to the best of our knowledge, the first evidence that overfitting is metric-dependent and a (baseline) solution for future work. Thereby, our solution provides hints at what could be the cause of this phenomenon  (e.g. sparseness or continuous values) and what this could mean for real-world AutoML systems (e.g., selecting the correct ensemble method per metric due to no free lunch).
> > > > >
> > > > > > "Shouldn't the proposed regularization help with balanced accuracy too?"
> > > > >
> > > > > If there is no overfitting, it should not help (see our initial comments on this). But maybe we can find a scenario in future work where there is overfitting for balanced accuracy and check if the normalization helps in this case (e.g. auto-sklearn 1 + ensemble).
> > > > >
> > > > > > "As a final comment, it would be great if the authors can point me to the autogluon paper that talks about it using GES. [...]"
> > > > >
> > > > > We understand your perspective since AutoGluon usually refers to GES only as "weighting" or "final stacking layer". Erickson et al. state this in their paper [R1]. Specifically, see Figure 2 and the last paragraph of Section 2.5, which states: "Our final stacking layer applies ensemble selection (Caruana et al., 2004) [...].". We refer to the work by Caruana et al. as greedy ensemble selection (GES) in our paper.
> > > > >
> > > > > &nbsp;
> > > > >
> > > > > Thank you for the thought-provoking and insightful discussions during this rebuttal.
> > > > >
> > > > > &nbsp;
> > > > >
> > > > > References
> > > > >
> > > > > * [R1] AutoGluon-Tabular: Robust and Accurate AutoML for Structured Data, https://arxiv.org/abs/2003.06505

---

> ### Author Response · Authors · 2023-04-24
> **Response to Review of Bj9B (Part 2/2)**
>
>
> > “Furthermore, the performance of CMA-ES with normalization seems to be significantly worse than that of CMA-ES with balanced accuracy. Does this mean that we would have to utilize different versions of CMA-ES ensembling for different metrics to get the best ensembling? This seems counter-intuitive.“
>
> Yes, that would be a takeaway for AutoML developers. It is very common for AutoML systems to change components based on the task and metric. For instance, AutoGluon switches parameters based on the provided metric, e.g. how to calibrate the final predictions or entier methods, e.g., not using stacking for binary classification. This would be the first time someone would select a different post hoc ensembling method. But considering that this field was not explored so far and that the no-free lunch theorem likely also holds, it does not seem counter-intuitive to us.
> Finally, the increase in performance provided by CMA-ES for balanced accuracy would certainly justify such a parameterization based on the metric. Additionally, we see balanced accuracy as representative for threshold-dependent metrics (F1, ..) and ROC AUC for threshold-independent metrics (log loss,...).
>
> > “On a related note, there is no discussion why the GES properties help CMA-ES with ROC-AUC but hurt CMA-ES with balanced accuracy. What might be happening in the ensembling that leads to such different behaviours with different metrics?”
>
> The reason is overfitting, i.e., the reason why we apply normalization. As mentioned in Section 2 and shown in Table 1, CMA-ES does overfit for ROC AUC but not for balanced accuracy. CMA-ES with normalization does not overfit while performing worse on validation and balanced accuracy – cases where there is no overfitting for CMA-ES anyways.
> As we failed to describe this explicitly, we will provide more details and discussion on this by extending the paragraph “Normalization to Combat Overfitting” of the results section in our updated manuscript.
>
> ### Use of AutoGluon base models
>
> We selected AutoGluon because it is likely the SOTA AutoML system (see the AutoML benchmark), uses GES, and one of its key concepts is a strong validation procedure (e.g., n-repeated k-fold cross-validation).
>
> Yes, the base model choices are pre-specified, and whether stacking is used or not. And what you are referring to with k-fold ensembling is what were are referring to as n-repeated k-fold cross-validation – as they build ensembles consisting of the fold models produced during cross-validation. HPO was not enabled.
>
> We agree that this differs from, for example, Auto-sklearn’s post hoc ensembling setup. But we think the differences are not significant enough, considering that both (Auto-Sklearn and AutoGluon) use GES and are affected by overfitting for gradient-free numerical optimization (proven for AutoGluon in our paper; stated for Auto-sklearn in their paper).
>
>
> ## Clarity
>
> We will update the manuscript with your feedback to improve clarity.
>
> We only have one minor immediate responses to clarify an open issue: An absolute rank of 1.5 occurs as CMA-ES-ExplicitGES is tied with GES. We only mentioned this in the results section. We will also explain this manifestation of the tie as part of the table. In detail, following common practices, we set the rank to the average of the ranks in case of a tie. As the methods here are tied for the first rank, resulting in taking rank 1 and 2; the rank is set to 1.5 for both.
>
> &nbsp;
>
> We hope our explanations and clarifications resolved your concerns, especially as you considered them to be fundamental limitations.
> Again, thank you for your detailed feedback and for sharing your ideas and thoughts for potential future work.

---

### Official Review · Reviewer_Ro2y · 2023-04-12

**Potential Impact On The Field Of Automl Rating:** 2
**Technical Quality And Correctness Rating:** 2
**Clarity Rating:** 2

**Summary Of Contributions:**

This paper proposes CMA-ES for post-hoc ensembling in AutoML. Therefore, first, the covariance matrix adaption evolution strategy (CMA-ES) is compared to Greedy Ensemble Selection (GES) as well as the single best model. Afterwards, CMA-ES is modified for three normalization techniques: softmax, softmax and implicit GES normalization, as well as softmax and explicit GES normalization. Those modifications are compared to each other as well as the base version of CMA-ES, stacking, and single best model. Furthermore, pseudo-discrete and sparseness of weight vectors are formalized, which is hypothesized to be properties that help to avoid overfitting.

The evaluation is done based on 10-fold cross-validations of 71 classification datasets from the AutoML benchmark of AutoGlueon, comparing the performance of balanced accuracy as well as ROC AUC both for binary and multi-class classification. The authors report that CMA-ES can prevent overfitting, and that it performs similary or better than GES.

**Actions Required To Increase Overall Recommendation:**

All in all the negative aspects described under correctness and clarity need to be addressed to convince me to raise my score. Additionally, I am interested in an argument why AutoSklearn2 has not been used instead of AutoSklearn(1)?

**Clarity:**

The structure of the paper is unusual and irritating. Most related work is already covered in the introduction, resulting in a very short related work. CMA-ES is only briefly introduced, then directly compared to GES, whereas the definition of GES is presented later in the paper under preliminaries (Section 3.1). At that point, the first evaluation has already been presented, mixing experimental setup, experiments and results. Results of a single experiment are covered within multiple sections (Sections 2.1 and 2.2), claiming significance before showing any results (table or plots) within that section (Section 2.1).

The formalization of the three normalization methods is not done well. If it is introduced mathematically, it has to be complete and correct, e.g., softmax is described w.r.t. the resulting weight vector, but it is completely unrelated to the original weight vector. Furthermore, the normalized improvement in Section 4 is described, but not mathematically introduced, which I would have expected to be precise. Additionally, the penalization with -10 is not further explained nor motivated.

The reason and motivation to formalize pseudo-discrete and sparse is not completely clear to me, i.e., why the authors hypothesize that both properties might help to avoid overfitting.

The appendix is not only used for additional information, but also for main statements of this paper, i.e., Figure 4 is referred within Section 3.4 but can be found only in the Appendix.

Within overall experiments in Section 4, stacking is included. There is no reason given why stacking has been used for comparison, neither why other standard ensembling techniques like bagging or boosting are not considered.

Moreover, the motivations and explanations of this paper are not well formulated.


**Overall Review:**

## Positive

- General idea of improving ensembling to prevent overfitting due to normalization is interesting.

## Negative (see comments in other fields)

- The correctness and quality of the presentation of experimental results is low and needs improvements. Furthermore, the conclusions need to be drawn with care.
- The clarity of the paper as well as motivation and precise descriptions needs to be improved.
- The contribution and novelty of the paper is very limited.


**Potential Impact On The Field Of Automl:**

The paper clearly states that post hoc ensembling in AutoML is addressed, which itself is an ongoing research direction within AutoML. As post hoc ensembling usually improves the performance of an AutoML tool, a better ensembling strategy is always of interest.

Nevertheless, the authors have bold statements about the achievements of the paper, which I don't interpret in the same way, maybe due to missing information on experiment results (see below).


**Review Confidence:**

4: You are confident in your assessment, but not absolutely certain. It is unlikely, but not impossible, that you did not understand some parts of the submission or that you are unfamiliar with some pieces of related work.

**Review Rating:**

4: Weak Reject: For instance, a paper with minor technical flaws, limited impact, and/or weak evaluation.

**Review Summary:**

The paper covers an interesting idea, but unfortunately not carefully written and analyzed. Especially the presentation of the experimental results is missing important information. Overall, I am wondering about the novelty of this paper due to the limited contribution. Additionally, I was wondering why AutoSklearn2 has not been used for comparison.

**Technical Quality And Correctness:**

Overall, the experimental results lack information. The captions of all tables state that there is a comparison of three methods, but only a single rank is given. I would expect to have a column per method (for each test/validation dataset) with concrete performance values, standard deviation (due to 10-fold cross-validation) as well as the rank of the according method, preferably also significance marks. Within the figures, half of the subfigure captions contain numbers within brackets without explanation, I guess those numbers refer to datasets. Within the text, some numbers from evaluation results are mentioned without having any table nor plot showing any of these (Section 3.4). The description of the results and the significance thereof does not match my interpretation of the result presentation.

---

> ### Author Response · Authors · 2023-04-24
> **Response to Review of Ro2y**
>
> Dear Reviewer Ro2y,
>
> Thank you for your feedback. The following addresses the negative aspects raised by you. As we want to start a potential discussion early, given the short timeframe of the rebuttal, we are sharing our comments now and are stating below where we plan to adjust the manuscript. The updated manuscript will then follow in the next few days.
>
> ### Technical Quality And Correctness
>
> > “The captions of all tables state that there is a comparison of three methods, but only a single rank is given.”
>
> The captions of the tables state that one method is compared to two other methods. Hence, only the results for one method are shown. We will provide tables with all rank changes and the details you requested in the updated manuscript.
>
> > “Within the text, some numbers from evaluation results are mentioned without having any table nor plot showing any of these (Section 3.4).”
>
> The section refers to Appendix E for a figure. An additional table for the mean number of base models was out of place, considering the page limit and small amount of numbers.
>
> > “The description of the results and the significance thereof does not match my interpretation of the result presentation.”
>
> We ask you to clarify this statement because the critical difference plots (Figures 1, 2, and 4) clearly show statistical significance using a Friedman test with the Nemeny post hoc test. How does your interpretation differ?
>
>
> ### Clarity
>
> > “Furthermore, the normalized improvement in Section 4 is described, but not mathematically introduced, which I would have expected to be precise. Additionally, the penalization with -10 is not further explained nor motivated.”:
>
> As we followed the AutoML benchmark, we did not deem it necessary to introduce it mathematically in our paper. We will add more details on this to explain the mathematical aspect. The penalization was motivated by PAR10 from algorithm selection. We only mentioned this in the code and failed to add it to the manuscript. We will update this part accordingly.
>
> > “The reason and motivation to formalize pseudo-discrete and sparse is not completely clear to me, i.e., why the authors hypothesize that both properties might help to avoid overfitting.”
>
> We partially answer this in Section 3.2. We will add more motivation by specifically stating that the good performance of GES for ROC AUC and its wide adaption (as indicated in the introduction) motivated us to look more into GES’s properties.
>
> > “The appendix is not only used for additional information, but also for main statements of this paper, i.e., Figure 4 is referred within Section 3.4 but can be found only in the Appendix.”
>
> Figure 4 provides additional detail but no main statement, as clarified in the first two sentences of Section 3.4. We did not select the method based on performance. Moreover, we could not include this figure in the main paper due to the page limit.
>
> > “Within overall experiments in Section 4, stacking is included. There is no reason given why stacking has been used for comparison, neither why other standard ensembling techniques like bagging or boosting are not considered.”
>
> We used Stacking since it relates to Auto-sklearn’s statement (Line 31-33). Moreover, stacking is used by H2o AutoML. Finally, stacking is a post hoc ensembling method, unlike bagging or boosting. Both require re-training of the base models and are therefore not considered post hoc ensembling but rather part of model selection by us. Moreover, neither has so far, to the best of our knowledge, been used for post hoc ensembling in any AutoML system.
>
> > “Additionally, I am interested in an argument why AutoSklearn2 has not been used instead of AutoSklearn(1)?”
>
> We are using AutoGluon. We ask you to clarify what you are referring to with this comment. In the introduction, we referred to Auto-Sklearn 1 because that is the version from the paper we are referring to (Auto-Sklearn 2’s paper did not comment on this again).
> Moreover, Auto-Sklearn 1 is the default when using Auto-Sklearn while Auto-Sklearn 2 is in an experimental phase.
>
>
> &nbsp;
>
> We are happy to follow up on any concerns you may have, especially the point regarding the significance analysis. We will also update the manuscript based on your remaining feedback.

---

> > ### Comment · Reviewer_Ro2y · 2023-04-24
> > **Highlight Changes**
> >
> > Dear Authors,
> >
> > Thanks for the fast reply. I don't have any concrete response to your message so far. Nevertheless, I would like to point out, that any changes you make to the paper should be highlighted, e.g. with a different color, so that reviewing the next version will be much easier.
> >
> > Thanks in advance!

---

> > ### Comment · Reviewer_Ro2y · 2023-05-01
> > **Response**
> >
> > Dear Authors,
> >
> > thanks for your detailed answers and the colored changes in the PDF. A lot of valuable information has been added to the paper. Nevertheless, besides the novelty, my main concern was about the structure and presentation of the content of your paper, which has not been reworked. Therefore, I am still not convinced to raise my score.

---

> > > ### Comment · Area_Chair_Vqah · 2023-05-02
> > > **Structure of the paper**
> > >
> > > Hi reviewer Ro2y,
> > >
> > > Are you sure that the structure of the paper is so bad that it justifies a score of 3? I see that the authors did not follow your advice to chance the structure, but I also see that your main argument was that the structure of the paper is different from what you view as the standard structure of a scientific paper. That alone is not enough. Maybe the structure is even better than the standard structure? That said, to be honest, I do not think that there is a unique best structure of a paper. In fact, my own papers have very different structures depending on the circumstances. If there is little previous work, you don't need a "previous work" section, for example. So I wonder if you could reconsider this case. Either, you make a convincing argument for the 3 (which is fine), or you take a fresh look at the paper, review the pros and cons, and give it a new grade? At the moment, it looks a little like you're annoyed by the fact that the authors did not change the structure and that's why you insist on your grade. I really value your opinion on the paper, but you should also consider the possibility that the authors are more experienced, have put more time into the paper, or have any other good reason to proceed as they do.
> > >
> > > All the best
> > > Your area chair

---

> > > > ### Comment · Reviewer_Ro2y · 2023-05-03
> > > > **Regarding structure of the paper**
> > > >
> > > > Dear area chair,
> > > >
> > > > I took a closer look at the paper again. For me, the structure of this paper is not just unusual, but also irritating. I agree that there is no standard paper structure every good paper has to follow and everyone has their own style of writing. While reading the paper I was wondering multiple times where information is coming from or if I missed something, but mostly it was due to the fact that those information was presented later in the paper (or even in the appendix).
> > > >
> > > > The main contributions of this paper are the application of CMA-ES to post-hoc ensembling, but more importantly, the 3 presented normalization methods to prevent overfitting. For me, the structure of the paper is not beneficial for the contributions, which is why I earlier (mistakenly) also criticized the novelty. Most importantly, the reader is extensively presented with CMA-ES post-hoc ensembling results, although that is not the main contribution. In contrast, the results of the main contribution, i.e. the normalization methods for CMA-ES for post-hoc ensembling are only very briefly discussed, but presented in the appendix. Therefore, in my opinion CMA-ES is over presented and the normalization methods are under presented and the paper is not self-contained as some of the main results are only presented in the appendix.
> > > >
> > > > Overall, for me, the paper was not easy to read and I had the impression that it was not carefully checked before submission, but due to mistakenly criticizing the novelty I will raise my score to a weak reject.

---

### Official Review · Reviewer_5CEF · 2023-04-12

**Potential Impact On The Field Of Automl Rating:** 2
**Technical Quality And Correctness Rating:** 3
**Clarity:** The paper is written clearly and is v…
**Clarity Rating:** 4
**Actions Required To Increase Overall Recommendation:** NA

**Summary Of Contributions:**

The paper discusses the use of post hoc ensembling in AutoML systems and proposes the use of a gradient-free numerical optimization method called Covariance Matrix Adaptation Evolution Strategy (CMA-ES), comparing its performance with the commonly used Greedy Ensemble Selection (GES) method. The authors apply these methods to AutoGluon on 71 classification datasets from the AutoML Benchmark and show that the choice of metric (balanced accuracy or ROC-AUC) affects whether GES or CMA-ES performs better. Specifically, for the metric balanced accuracy, CMA-ES outperforms GES, while for ROC AUC, GES outperforms CMA-ES due to overfitting. Additionally, the authors propose a method to avoid overfitting for CMA-ES, which involves a novel normalization method inspired by GES's implicit constraints during optimization. This method allows CMA-ES to perform as well as GES while keeping the ensemble size small.

**Overall Review:**

Strengths:

-The paper presents a detailed analysis of the performance of two post hoc ensembling methods, GES and CMA-ES, and compares their performance on 71 classification datasets from the AutoML Benchmark. This comparison provides valuable empirical contribution to the field of AutoML.
-The authors propose a novel way to avoid overfitting in post hoc ensembling using CMA-ES, which involves a normalization method inspired by GES's implicit constraints during optimization. This method can improve the performance of CMA-ES on the ROC AUC metric, which is known to be prone to overfitting.
-The paper provides insights into the effect of different validation data qualities on post hoc ensembling performance, highlighting the importance of using high-quality validation data.

Weakness:

-The paper does not apply the methods to other AutoML systems and limits the analysis to AutoGluon. Extending the evaluation to other AutoML systems would provide more generalizable insights.




**Potential Impact On The Field Of Automl:**

The paper focuses on the topic of post hoc ensembling in AutoML, which has been relatively underexplored. It is the first to apply the CMA-ES technique to directly optimize the weights of an ensemble, and it also proposes a novel way of dealing with overfitting in post hoc ensembling.

**Review Confidence:**

4: You are confident in your assessment, but not absolutely certain. It is unlikely, but not impossible, that you did not understand some parts of the submission or that you are unfamiliar with some pieces of related work.

**Review Rating:**

7: Weak Accept: Technically sound paper with moderate-to-high impact and strong evaluation, with perhaps some minor flaws.

**Review Summary:**

The paper provides a detailed comparison of two post hoc ensembling methods, GES and CMA-ES, and proposes a novel approach to avoid overfitting in post hoc ensembling using CMA-ES. Additionally, the paper provides empirical evidence to highlight the importance of using high-quality validation data for better results in post hoc ensembling. While there are some limitations to the paper, the insights provided are valuable for practitioners in the field.

**Technical Quality And Correctness:**

The paper distinguishes between systems that use high-quality validation data and those that do not, such as autosklearn using only a 33% holdout. The authors conclude that overfitting does not depend on the metric when tested on systems that use high-quality validation data. It would be interesting to see how the results vary with different folds for methods that use cross-validation and with different test percentages for those that use holdout.

Furthermore, it is worth noting that AutoGluon and Auto-Sklearn use different components and pipeline architecture. It would be interesting to investigate whether Auto-Sklearn using high-quality validation data can also support the claims made in this paper.

---

> ### Author Response · Authors · 2023-04-24
> **Clarification Regarding a Statement in Technical Quality And Correctness**
>
> Dear Reviewer 5CEF,
>
> Thank you for your feedback. We request a short clarification regarding your review:
>
> You stated under Technical Quality And Correctness: “The authors conclude that overfitting does not depend on the metric when tested on systems that use high-quality validation data.” – The “not” contradicts your previous statements. Was this intended as feedback or an error?

---

> > ### Comment · Reviewer_5CEF · 2023-04-24
> > **Clarification**
> >
> > Dear Authors,
> >
> > I apologize for the error. The "not" was unintended. I appreciate your attention to detail and for pointing out the inconsistency.

---

### Author Response · Authors · 2023-04-25
**Updated Manuscript Available**

Dear Reviewers,

The updated manuscript is available. *All changes are colored green*.

In detail, we changed the following major points, among other minor things:

* We explained our choice of CMA-ES in the introduction.
* We explicitly state our motivation for analyzing GES in Section 4.
* We fully formalized the definitions of our normalization methods (Section 4.3). This includes an algorithmic description of our proposed approach.
* We motivated including stacking in our final comparison.
* We provide an additional discussion in the result section on the performance difference between balanced accuracy and ROC AUC.
* We included an overview of the rank change from validation to test data for all compared methods in the Appendix.
* We extended the Appendix on the comparison of normalization method w.r.t. an ablation study and a table for the average ensemble sizes.
* We provide a detailed description of the used statistical test and normalized improvement metric.
* We added tables showing the mean score and standard deviation per dataset in the Appendix.

We believe these changes address the concerns raised by you.
Therefore, we politely ask you to verify if your concerns have been resolved and reconsider your rating.

Thank you for your time and valuable feedback.
We are happy to participate in any discussions and to resolve additional concerns.